# Molecular basis for recognition and deubiquitination of 40S ribosomes by Otu2

Ken Ikeuchi [1], Nives Ivic[2], Robert Buschauer [1], Jingdong Cheng [1,3], Thomas Fröhlich [4], Yoshitaka Matsuo [5], Otto Berninghausen [1], Toshifumi Inada[5], Thomas Becker [1] ✉ & Roland Beckmann [1] ✉

In actively translating 80S ribosomes the ribosomal protein eS7 of the 40S subunit is monoubiquitinated by the E3 ligase Not4 and deubiquitinated by Otu2 upon ribosomal subunit recycling. Despite its importance for translation efficiency the exact role and structural basis for this translational reset is poorly understood. Here, structural analysis by cryo-electron microscopy of native and reconstituted Otu2-bound ribosomal complexes reveals that Otu2 engages 40S subunits mainly between ribosome recycling and initiation stages. Otu2 binds to several sites on the intersubunit surface of the 40S that are not occupied by any other 40S-binding factors. This binding mode explains the discrimination against 80S ribosomes via the largely helical N-terminal domain of Otu2 as well as the specificity for mono-ubiquitinated eS7 on 40S. Collectively, this study reveals mechanistic insights into the Otu2-driven deubiquitination steps for translational reset during ribosome recycling/(re)initiation.

The addition of ubiquitin to lysine residues of a protein or to ubiquitin itself is a very common post-translational modification. It creates multi-faceted cellular signals, known as the 'ubiquitin code' that can lead to a multitude of possible cellular consequences such as protein degradation, signaling and trafficking, epigenetic regulation, cell cycle control, and many more[1]. Commonly, ubiquitin is added via sequential action of three enzyme classes, E1 ubiquitin-activating enzymes, E2 ubiquitin-conjugating enzymes and E3 ubiquitin ligases. The effects of ubiquitination are counteracted by deubiquitinating enzymes (DUBs), that are able to remove (poly)ubiquitin chains[2].

Recently, important roles of ubiquitination and deubiquitination have emerged in the context of eukaryotic translation. For example, small subunit (SSU) proteins uS3, uS10 and eS10 are ubiquitinated following ribosome collision after prolonged stalling of translation[3–11]. This can trigger mRNA surveillance and ribosome-associated quality control (RQC) pathways, during which stalled ribosomes are

dissociated into subunits and the mRNA as well as the arrest peptide are degraded[11–20]. uS3 and uS5 are also ubiquitinated under several types of translational stress[21–25] and sequential ubiquitination of uS3 can promote the 18S non-functional ribosomal RNA decay (NRD) pathway[26,27]. Another important target for regulation of gene expression is eS7. In yeast, eS7 is monoubiquitinated by the E3 ligase Not4[28], a component of the Ccr4-Not complex and master regulator of gene expression[29]. Ubiquitinated eS7 was first found mainly on polysomes[28] but not on 40S subunits, and it was shown to prevent protein aggregation[28]. Later studies suggest that ubiquitinated eS7 is present on 80S ribosomes and that presence of monoubiquitinated eS7 in 80S is important for translational efficiency[30]. Moreover, it was demonstrated to be required for mRNA homeostasis by serving as a prerequisite of Ccr4-Not's role in triggering degradation of non-optimal mRNAs[31,32]. In this context, eS7-monoubiquitination is required to recruit Not5 to the polysomes[28], where it is sensing the empty E-site of

[1]Department of Biochemistry, Gene Center, Feodor-Lynen-Str. 25, University of Munich, 81377 Munich, Germany. [2]Division of Physical Chemistry, Rudjer Boskovic Institute, Bijenicka cesta 54, 10000 Zagreb, Croatia. [3]Institutes of biomedical science, Shanghai Key Laboratory of Medical Epigenetics, International Co-laboratory of Medical Epigenetics and Metabolism (Ministry of Science and Technology), Fudan university, Dong'an Road 131, 200032 Shanghai, China. [4]LAFUGA, Laboratory for Functional Genome Analysis, Gene Center, Feodor-Lynen-Str. 25, University of Munich, 81377 Munich, Germany. [5]Division of RNA and Gene Regulation, Institute of Medical Science, The University of Tokyo, Minato-ku 108-8639, Japan. ✉e-mail: becker@genzentrum.lmu.de; beckmann@genzentrum.lmu.de

slow ribosomes[31]. Further functions of eS7-monoubiquitination were described in context with selective mRNA translation under ER stress conditions[33] and to prevent translation of poly-arginine codons[32]. Finally, in yeast, eS7 can be polyubiquitinated by the E3 ligase Hel2 to trigger non-canonical No-Go mRNA decay (NGD) in a bypass quality control pathway[4].

eS7 poly- and monoubiquitination is antagonized by deubiquitinating enzymes Ubp3 (USP10 in human) and Otu2[30,33], While Ubp3 preserves the eS7 monoubiquitination, Otu2 was shown to specifically remove the monoubiquitin attached to Lys83 of eS7[30]. This step exclusively occurs on 40S ribosomal subunits and was suggested to happen during recycling of mRNA and tRNA from 40S after translation termination[30].

However, several questions remain open. Given that efficiently translating ribosomes require monoubiquitination of eS7[28], Otu2 activity must be limited to free 40S subunit and discriminate against 80S ribosomes. Yet, eS7 is equally accessible in 40S and 80S ribosomes and it is therefore not clear how 40S specificity is achieved. While it was previously shown that Otu2 engages recycled 40S and 43S pre-initiation complexes[30], no structural information on this interaction is available so far.

In this study we present a cryo-EM inventory of Otu2-containing 40S ribosomal particles, that in combination with mass spectrometric analysis, in vitro reconstitution and biochemical assays sheds new light on the activity of Otu2 during translation. We find that Otu2 mainly associated with 43S pre-initiation complexes (43S-PIC) and 48S initiation complexes (48S-IC), with both populations also associating with ABCE1. In addition, we find a small amount of pre-40S assembly intermediates which bind to Otu2 together with other 40S biogenesis factors. In all populations, we identify Otu2 bound to specific sites of the 40S intersubunit surface that have not been described before for other 40S-binding factors. 40S specificity is thereby established via the largely helical N-terminal domain (NTD) of Otu2, that is binding to sites that are masked and therefore inaccessible in the 80S ribosome. The C-terminal domain containing the (catalytically dead) deubiquitinating OTU-domain together with ubiquitin are visible and stably positioned in all classes containing mono-ubiquitinated eS7, i.e., all except the pre-40S, where the OTU domain was delocalized. Combined with a crystal structure of the OTU domain, we establish the molecular basis for recognition of eS7 with Lys83-linked monoubiquitin, explaining the unusual substrate specificity of Otu2 compared to other members of its subfamily of deubiquitinating enzymes. Collectively, we establish Otu2 as a factor that can generally bind to 40S during cytoplasmic ribosome biogenesis and throughout all stages between ribosome recycling and start-codon recognition, but becomes a highly specific enzyme for translational reset during ribosome recycling/(re)initiation by removing mono-ubiquitin present on eS7 in a 40S ribosomal subunit.

## Results

### Otu2 associates with 40S recycling and initiation complexes

To explore the role of ribosome-bound Otu2 and Ubp3, and to identify additional factors associated with the respective complexes, we performed a quantitative mass spectrometry-based analysis. To that end, we affinity purified Otu2-FTpA (FLAG-TEV-proteinA) and Ubp3-FTpA associated complexes from yeast cells (Fig. 1a, b) and analyzed them by label-free mass spectrometry (LC-MS/MS) (Fig. 1c, see also Source Data). Statistical analysis of the LC-MS/MS data revealed that, in the Ubp3-bound fraction, primarily its co-factor Bre5 (G3BP1 in human)[34] as well as large ribosomal subunit (LSU) proteins were enriched, in agreement with a suggested function of Ubp3 on 80S ribosomes[30,33,35–38]. In addition, we found factors involved in ER to Golgi trafficking which is in agreement with a known function of Ubp3 independent of translation[34,39]. In contrast, in the Otu2-associated fraction we found enrichment of SSU proteins and of factors involved

in translation termination, ribosome and mRNA recycling (ABCE1; Rli1 in yeast, release factors, Tma64; eIF2D in human) as well as proteins involved in SSU biogenesis (Fig. 1c, Source Data). In addition, translation initiation factors (subunits of the eIF2 and eIF3 complexes) were enriched in the wash fraction after micrococcal S7 nuclease treatment, that contained mRNA-mediated or less stably associated factors (Fig. 1d). Independent of its role in translation, Otu2 may play an additional role in 40S biogenesis which was already shown for OTUD6B, the human homolog of Otu2[40]. Although a functional role of eS7 (de)ubiquitination during pre-40S formation in yeast has not yet been described, the enrichment of biogenesis factors in Otu2 complexes supports this idea. The observation of strong enrichment of 40S ribosomal proteins clearly indicates that in contrast to 80S associating Ubp3, Otu2 binds with high preference to 40S subunits. Based on the enrichment of the recycling factors ABCE1 and Tma64, we conclude that Otu2 association can already occur early during recycling, immediately after ABCE1-induced dissociation of the 60S subunit[41–45] and during dissociation of the mRNA facilitated by factors like Tma20/22 (MCT-1 and DENR in human) and Tma64 (eIF2D or Ligatin in human)[42,46–48]. This is consistent with earlier biochemical findings that the impairment of Otu2 activity results in a reduced efficiency of mRNA recycling[30]. The presence of Otu2 on the recycled 40S can last into the phase of 43S pre-initiation complex formation as indicated by the presence of eIF2 and eIF3 subunits. This may hint at an additional function of Otu2 after mRNA recycling during (re-)initiation.

### Otu2 binds to 40S for deubiquitination of mono-ubiquitinated eS7

Next, we further analyzed ribosome association and eS7 deubiquitination activity in yeast cell lysates. Here, we used either wild type (wt) Otu2 or Otu2 mutated in residues forming the catalytic triad present in members of the OTU family of deubiquitinases which are cysteine peptidases[2,49]. In Otu2's OTU domain, we predicted the catalytic triad to be formed by Cys178, His300 and Asn302, based on sequence alignments (Supplementary Fig. 1) and a C178S mutation was already shown before to be catalytically inactive[30].

We expressed C-terminally 3x FLAG-tagged Otu2, otu2-C178S and otu2-H300A from plasmids under the endogenous promoter in yeast strains with otu2Δ or otu2Δubp3Δ background harboring shuffled hemagglutinin (HA)-tagged eS7A to monitor ubiquitination. Western blot analysis of lysates subjected to sucrose density gradient fractionation confirmed that both active and inactive Otu2 are associated with 40S subunits and that deubiquitination of eS7 on 40S is dependent on the catalytic activity of Otu2 (Supplementary Fig. 2a, 2b). Moreover, catalytic inactivity is accompanied by a decrease of polysomes, emphasizing the proposed importance of Otu2 function for efficient translation (Supplementary Fig. 2c). This is further supported by the earlier observation, that the synthetic growth defect of otu2Δubp3Δ cells[30] is even more pronounced in the presence of translation elongation inhibitors or at higher temperature (Supplementary Fig. 2d).

### Cryo-EM structures of native Otu2-bound 40S complexes

To obtain a structural inventory of native Otu2-bound 40S populations, we initially chose a shotgun cryo-EM approach similar to a previous study focusing on native ABCE1-bound ribosomal complexes[50]. After overexpressing the catalytically inactive otu2-C178S mutant in yeast, we harvested the crude 40S/43S/48S peak from a preparative sucrose density gradient of cell lysate without further purification to avoid loss of the loosely associated initiation factors (see Fig. 1d). Classification revealed three main sets of Otu2-bound 40S particles, representing eS7-monoubiqitinated 43S pre-initiation (Otu2-Ub-43S-PIC) and (partial) 48S initiation complexes (Otu2-Ub-48S-IC) as well as unmodified pre-40S particles (Otu2-pre-40S) (Supplementary Fig. 3a). Extra density for Otu2 was visible at mono-ubiquitinated eS7 and on

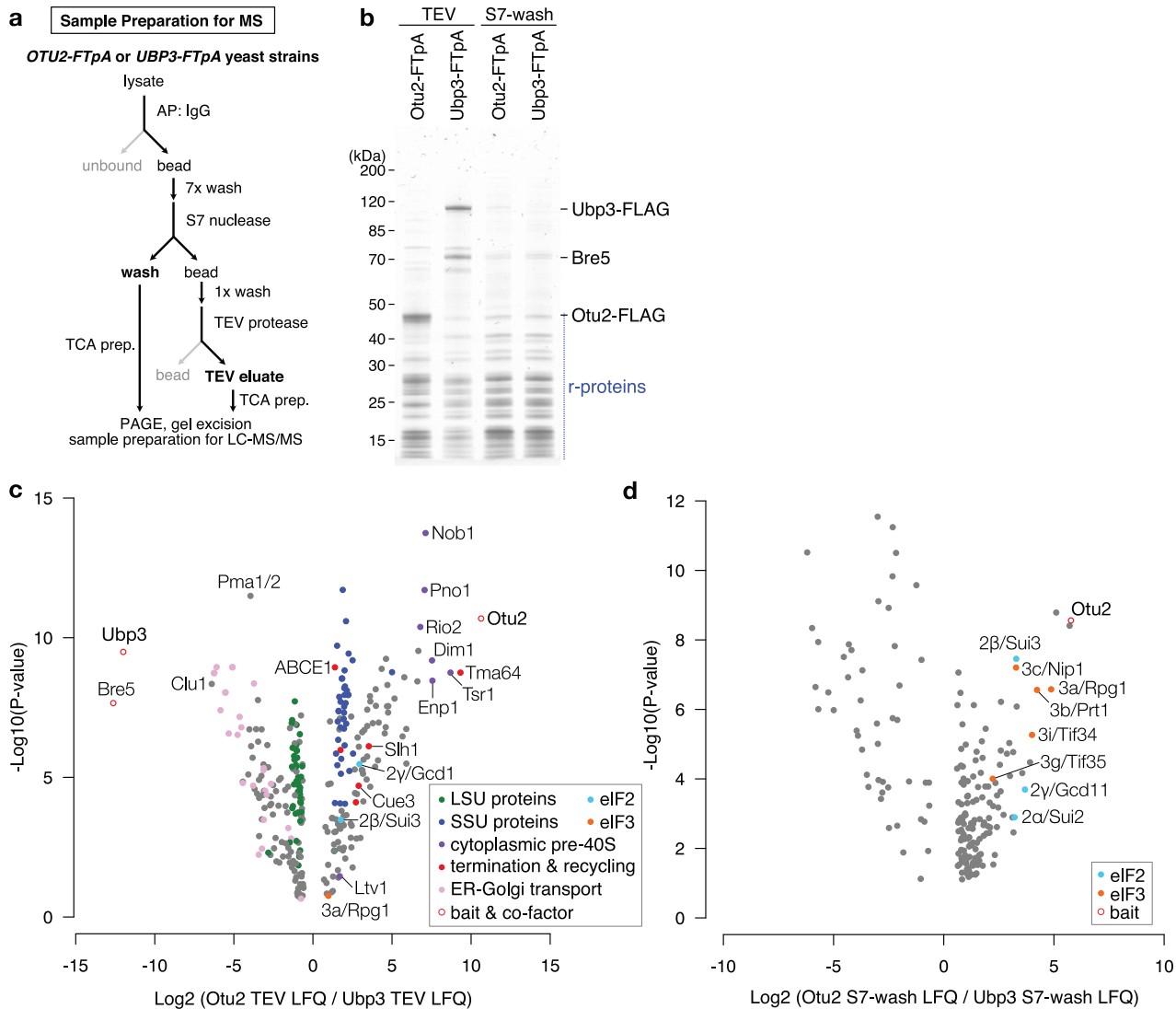

**Fig. 1 | Quantitative mass spectrometry-based analysis of ribosome-bound Otu2 versus Ubp3. a** Scheme outlining the sample preparation for mass spectrometry. MS mass spectrometry, AP affinity purification, TCA trichloroacetic acid, TEV tobacco etch virus. **b** Nu-PAGE gel of affinity purified fractions by Otu2-FTpA and Ubp3-FTpA. r-proteins; ribosomal proteins. We obtained essentially the same results in at least three independent experiments. The original gel is provided in the Source data file. **c** Volcano plot showing the fold change (log2 LFQ Otu2-TEV/LFQ Ubp3-TEV) on the x-axis and the P-value distribution on the y-axis for the proteins identified in the affinity purifications. Missing values were imputed from a normal distribution (width, 0.3; down-shift, 1.8). To test for differentially abundant proteins, a two-sided T-test was employed. Multiple testing correction was performed with a permutation-based FDR estimation (FDR < 0.05). q-values ≤ 0.05 with log2-fold changes < −0.6 and > 0.6 were considered as statistically significant. Each circle indicates an identified protein. Hits were grouped in categories and color coded as indicated in the panel. Gray dots represent unspecific or unrelated hits. Enriched proteins from the Otu2 purification can be found on the right side, de-enriched ones on the left side. LFQ label-free quantification, SSU and LSU small and large ribosomal subunit, ER endoplasmic reticulum. **d** Volcano plots as in **c** for wash samples after S7 nuclease treatment highlighting enriched initiation factors. Source Data for (**c**) and (**d**) are provided in the Source data file, that contains the complete inventory.

the intersubunit side of the 40S body (see below). These densities could be assigned to the Otu2 OTU domain (at eS7) and to the Otu2 N-terminal domain (Fig. 2) (see below).

The initiation complexes were distinguished by the presence of eIF1, eIF1A, eIF3 and eIF3j for the 43S-PIC (Fig. 2a), and additional density for the ternary complex (TC; consisting of trimeric eIF2αβγ complex, initiator methionyl tRNA (tRNAi) and GTP), the N-terminal domain of eIF5 (eIF5-NTD) and lack of eIF1 and eIF3j for the 48S-IC (Fig. 2b). Both 43S and 48S classes contained sub-classes with ABCE1 bound as described before[50]. The pre-40S class contained a set of well-described biogenesis factors, such as Tsr1, Dim1, Rio2, Enp1-Ltv1, Pno1, and Nob1(Fig. 2c), thus resembling a stable pre-40S particle prior to trimming of 20S pre-rRNA at the 3′-end.

In addition to the native sample, we generated an Otu2-40S complex sample in vitro for cryo-EM analysis. To that end, we reconstituted an Otu2-eS7-Ub-40S complex using an in vitro ubiquitination system (Supplementary Fig. 4a). Herein, purified 80S ribosomes were monoubiquitinated with purified Not4 E3 ligase, Ubc4 (E2), Uba1 (E1), ubiquitin and ATP to obtain 80S with monoubiquitinated eS7 (eS7-Ub-80S; Supplementary Fig. 4b). These ribosomes were split into subunits (Supplementary Fig. 4c) and purified eS7-Ub-40S were incubated with purified recombinant Otu2 to ensure efficient deubiquitination activity in this system (Supplementary Fig. 4d). After observing complete eS7-deubiquitination by recombinant purified wild-type (wt) Otu2, we used the eS7-Ub-40S for reconstitution with a 5x molar excess of purified catalytically inactive otu2-C178S to form stable complexes

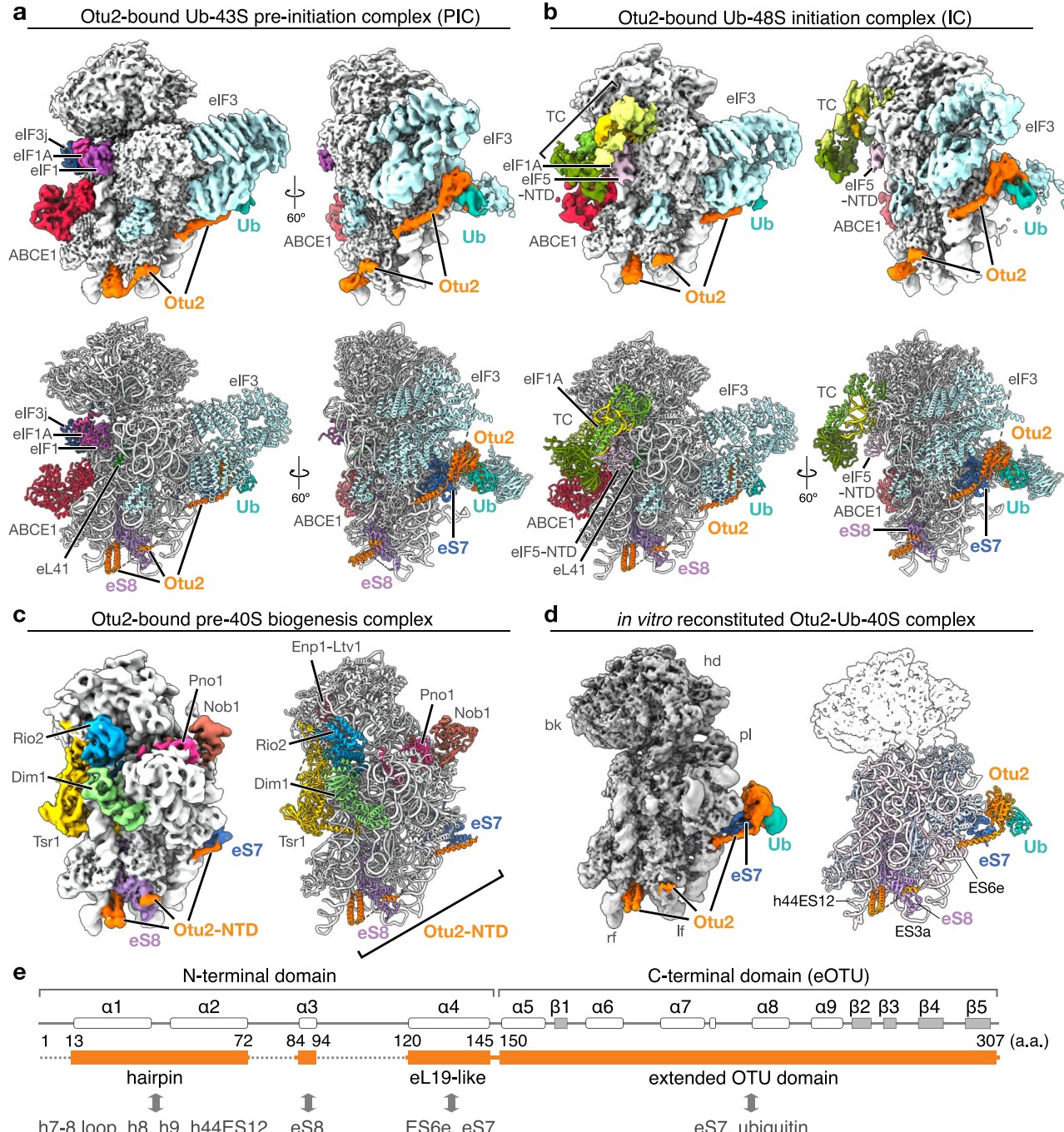

**Fig. 2 | Cryo-EM structures of native Otu2-40S complexes. a, b** Cryo-EM density maps (top) and molecular models (bottom) of the Otu2-bound and eS7 mono-ubiquitinated 43S-PIC complex (**a**) and 48S-IC (**b**). **c, d** Cryo-EM density maps (left) and molecular models (right) of the Otu2-bound late pre-40S complex (unmodified eS7) and the in vitro reconstituted Otu2-Ub-40S complex (**d**). In **d**, only the model for the 40S body is shown and the 40S head is displayed as transparent density. The map in (**d**) is displayed as composite map after multi-body refinement (see Supplementary Figs. 3a and 5) and low-pass filtered according to local resolution (see Supplementary Fig. 5). TC ternary complex consisting of trimeric eIF2αβγ complex, initiator methionyl tRNA (tRNAi) and GTP; ISS intersubunit surface, SS solvent surface, hd head, bk beak, pl platform, rf right foot, lf left foot, Ub ubiquitin. **e** Schematic illustration of Otu2 secondary structure and domains. Structurally visible parts are highlighted as orange boxes and interaction sites with the 40S are indicated.

(Supplementary Fig. 4e) and determined the structure by cryo-EM (Fig. 2d and Supplementary Figs. 4f and 5).

All described structures were refined to a high average resolution of 3.0 Å for Otu2-Ub-43S-PIC, 3.3 Å for Otu2-Ub-48S-IC, 3.8 Å for Otu2-pre-40S, and 3.0/3.2 Å for the two bodies of Otu2-Ub-40S (Supplementary Fig. 5). To improve local resolution of more flexible regions, focused 3D variability analysis followed by local refinement in CryoSPARC[51,52] or multi-body refinement in RELION[53,54] was performed.

To obtain the highest possible resolution for Otu2 all particles showing extra density at eS7 (Otu2-C; OTU domain) or extra density for the N-terminus (Otu2-N; for definition see Fig. 2e) were locally refined together, yielding local resolutions of 2.9 Å (for Otu2-N) and 3.0 Å (for Otu2-C), respectively (Supplementary Figs. 3a, 4f, and 5). At this resolution, molecular models for Otu2-bound 43S/48S, pre-40S and 40S could be built based on a AlphaFold v2.0 (AF2)[55] model for Otu2 (Supplementary Fig. 6) as well as based on previous models for the

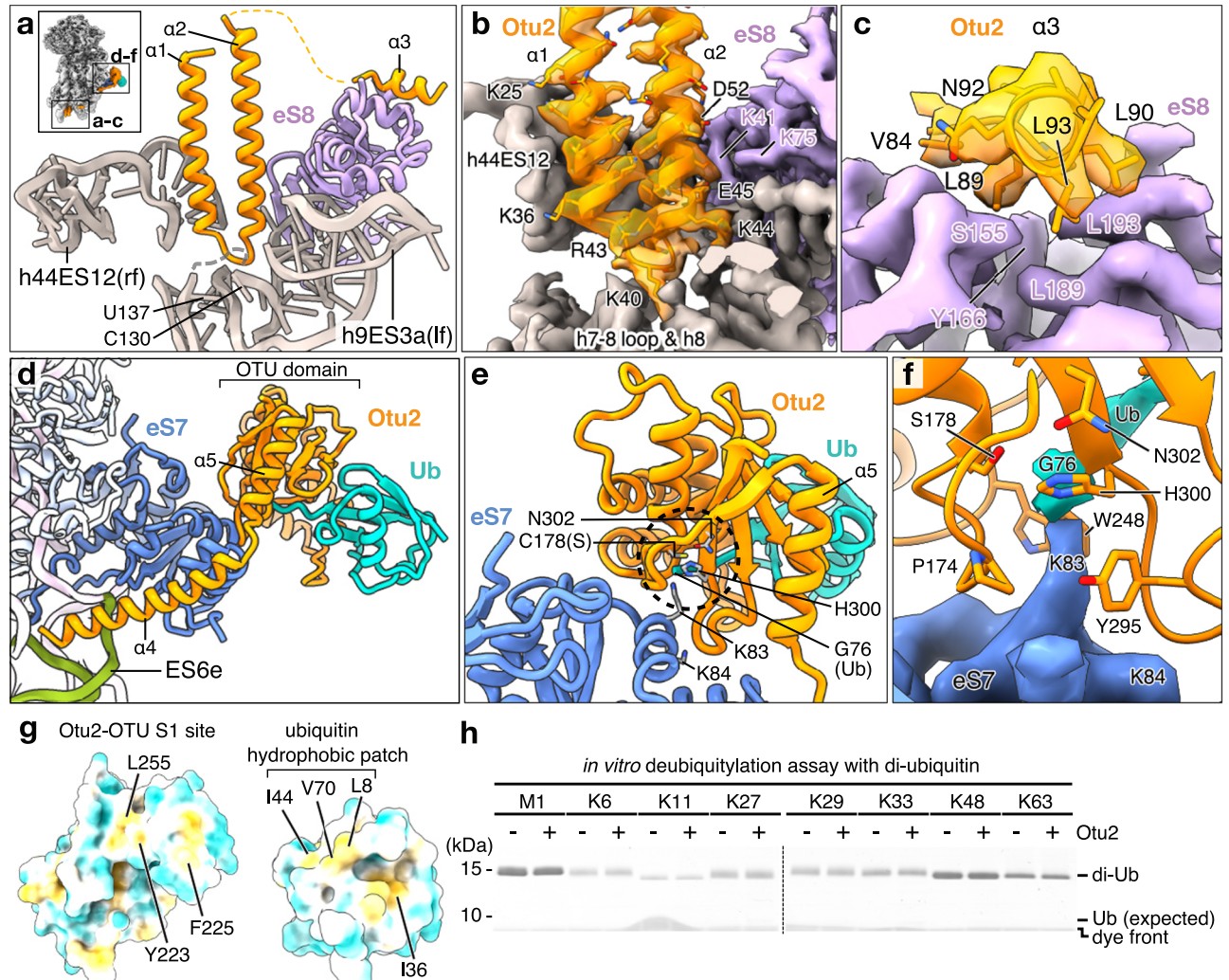

**Fig. 3 | Structure and specificity of Otu2 binding to eS7-monoubiquitinated 40S. a** molecular model of the Otu2 N-terminal hairpin (α1- α2) bound in a pocket formed by h44 (ES12), the loop between h7 and h8, h9 (ES3a) and eS8 as well as of Otu2 α3 bound to eS8. A thumbnail in the top left corner indicates the location of zoomed regions within the Otu2-40S complex. rf right foot, lf left foot. **b** view as in **a**, showing density for the Otu2 α1- α2 hairpin (transparent orange with fitted model) in its binding pocket (solid map). Interacting residues are labeled. **c** side view focusing on the Otu2 α3-eS8 interaction site. **d** molecular model of the extended Otu2 OTU domain and α4 bound to Lys83-monoubiquitinated eS7 and ES6e. **e** zoom view focusing on the model for the active site of OTU domain targeting the ubiquitinated-eS7-Lys83 residue. The dotted circle shows the protease

site with the catalytic triad (Cys178, His300 and Asn302) of Otu2 and its substrate, ubiquitinated eS7-Lys83. The minor ubiquitination site eS7-Lys84 is also shown. Ub ubiquitin. **f** zoom on the eS7-Lys83 ubiquitination site (eS7 and the C-terminus of monoubiquitin shown as electron potential map) contacted by Pro174 (in loop β1-α6), Trp248 (loop α8-α9) and Tyr295 (loop β4-β5) of the OTU domain (shown as model). **g** "Open book" view displaying the hydrophobic contact interface between the Otu2 OTU-S1 site and ubiquitin. Hydrophobic regions are shown in yellow, hydrophilic in cyan. **h** SDS gel of an in vitro deubiquitination assay using purified Otu2 and di-ubiquitin with indicated linkage types. We obtained essentially the same results in at least three independent experiments. The original gel image for (**h**) is provided in the Source Data file. Ub ubiquitin.

respective complexes (Fig. 2, Supplementary Figs. 3b, c, 6a, b, and 7, and Supplementary Table 1)[50,56,57].

## Structure of Otu2 bound to 40S

In all electron potential maps (densities) we observed extra density for Otu2 at sites that so far have not been annotated as interaction sites for other known 40S-binding factors. We observed parts of the Otu2-specific N-terminus (1–149) which winds along the lower part of the 40S body on the intersubunit side (ISS) connecting the 'foot' structure (formed by the lower parts of rRNA helix h44 that extends into expansion segment ES12) with the OTU domain (Fig. 2 and Supplementary Fig. 7a). We were able to build a molecular model for the four α-helices (α1-α4) based on a predicted model generated by AF2 (Figs. 2 and 3a–c and Supplementary Figs. 3b, 6a, b, and 7b): α1 and α2 form a hairpin that is inserted into a deep rRNA pocket formed by residues of

h7-h8 loop, h9/ES3a and the lower end of h44/ES12 (Fig. 3a, b and Supplementary Figs. 7a, b). The hairpin is flexibly linked to the short α3 helix of Otu2 (also predicted by AF2) that binds on the surface of eS8 (Fig. 3c). A longer flexible linker connects α3 with the long α4, that inserts into the major groove of the rRNA expansion segment ES6e and projects further towards eS7, where it connects to the OTU domain of Otu2 (167–305) (Fig. 3d–f and Supplementary Fig. 7c, d). This domain as well as monoubiquitin adjacent to eS7 was visible in all structures except the pre-40S classes (Fig. 2a–c). Similar to OTUD1 and OTUD3, the OTU domain of Otu2 is extended on the N-terminal side by an α-helix (α5; 150–166, Supplementary Figs. 1b and 6) that contacted eS7.

Since local resolution of the OTU domain was not sufficient to build a de novo molecular model, we determined the structure of the N-terminally extended Otu2-OTU domain comprising residues 150–307 by X-ray crystallography (Supplementary Table 2 and

Supplementary Fig. 6d). This structure was highly similar to the OTU domain structure predicted by AF2 (Supplementary Fig. 6e, f; RMSD of 0.367) except for loops contacting Lys83-monoubiquitinated eS7 (see below) and fitted well into our electron potential map. Here, we could also unambiguously identify ubiquitin based on secondary structure (Supplementary Figs. 3c, 5k, and 7c, e) enabling us to obtain an overall model of eS7-mono-Ub bound Otu2.

The structure of the OTU domain of Otu2 showed the typical OTU domain fold and is most similar to human OTUD1 and OTUD3 domains (Supplementary Figs. 1b and 6h, i), both of which have the N-terminal α-helix of the catalytic domain (α5 in Otu2; Leu150-Lys165; also called the S1' helix), that is suggested to contribute to the ubiquitin linkage specificity and/or substrate specificity[49,58]. Examination of the position of the Otu2 OTU domain in our cryo-EM map revealed that the conserved core of the domain forms numerous contacts to Lys83-ubiquitinated eS7 (Fig. 3d–3f and Supplementary Fig. 7c, d). Additional eS7 contacts are formed by the region linking the S1'-helix (α5) with α4 (140–151) (Fig. 3d). Although the local resolution was lower due to flexibility (4.0–7.0 Å), we observed a clear density for ubiquitin bound to the Otu2 S1 site (Supplementary Figs. 3c, 5k, and 7c, e). Like in other Ub-binding proteins and OTU domain containing proteins, Otu2 binds to ubiquitin via a conserved hydrophobic region in the S1 site of the OTU domain (Tyr223, Phe225, Leu255) recognizing a corresponding hydrophobic patch in ubiquitin (Leu8, Ile44, Val70), as described before for various types of ubiquitin-binding proteins including the OTU family of DUBs[2,49,58–61] (Fig. 3g). The position of the Otu2-bound ubiquitin on the 40S subunit localizes its C-terminal Gly76 in close proximity to the modified Lys83 (but not Lys84) of eS7, which is consistent with a Lys83-linked eS7-ubiquitin substrate complex engaged by the catalytically inactive Otu2[4,30], (Fig. 3e, f). Here, residues in loops β1-α6 (Pro174), α8-α9 (Trp248) and β4-β5 (Tyr295) - all adjacent to the catalytic triad (Cys178, His300, and Asn302) - establish specific contacts to eS7 (Fig. 3f and Supplementary Fig. 7d), suggesting that the activity of Otu2 is highly specific for eS7-Lys83 linked ubiquitin. Indeed, in vitro deubiquitination assays showed no activity of Otu2 on di-ubiquitins, irrespective of their linkage type (Fig. 3h).

### Otu2 in context of initiation and 40S maturation complexes

As shown in Fig. 2, in our native sample we observed Otu2 bound to 43S-PIC, 48S-IC and late pre-40S particles. For the structure of the 43S-PIC, we observed similar heterogeneity in occupancy and conformation of initiation factors as described before[50]. The most stable class showed a 40S subunit in a closed-latch conformation with eIF1, eIF1A and eIF3, eIF3j (Hcr1 in yeast) and ABCE1 (Rli1 in yeast) in positions previously described[50,62–64] (Fig. 2a). The 48S-IC contained eIF1, eIF3, the NTD of eIF5, ABCE1 as well as initiator tRNA (tRNA_i) as observed before[62,63,65,66] with tRNA_i in the P_IN conformation, but contrary to our previous study[50], also density for the eIF2 complex was present in the most stable class (Fig. 2b).

The pre-40S particle (Fig. 2c) resembles a late-stage cytoplasmic assembly intermediate prior to cleavage of 20S rRNA at the 3'-end by endoribonuclease Nob1. It contains biogenesis factors Tsr1, Rio2, Dim1 at the intersubunit side of the 40S, Pno1 at the platform, and Enp1-Ltv1 at the beak, all in positions essentially as described before[56,57,67–69]. In addition, our particle contains Nob1 in its inactive conformation[56] as also observed in the recently identified "Rio2-C" state[69]. Interestingly, unlike in all other observed yeast pre-40S particles, we observed uS10 and RACK1 already incorporated, yet corresponding density is relatively weak. In addition, uS3 was only partially accommodated as expected.

Notably, in all native complexes, we observed no direct interaction of Otu2 with any other non-ribosomal factor present in our structures. Yet, we found that the OTU domain is located in close vicinity to eIF3c on the back-side of the 40S (Supplementary Fig. 7e), a location from which it could influence assembly of

initiation complexes by only small conformational changes of the OTU domain.

In the pre-40S complex, only the N-terminal domain of Otu2 is visible (Fig. 2c), whereas the OTU domain is delocalized and also eS7-mono-Ub is not present. To test for eS7-ubiquitination in pre-40S complexes, we purified pre-40S via tagged Rio2 in yeast shuffle strains harboring HA-tagged eS7 in presence of wt Otu2 or otu2-C178S. While overall levels of eS7 mono-ubiquitination were slightly elevated in whole cell lysates with the catalytically dead Otu2 (Supplementary Fig. 8a), we did not observe any ubiquitinated eS7 in the assembly intermediates after the affinity purification (Supplementary Fig. 8b). This indicated that these pre-40S complexes, and likely also the pre-40S observed in our native structure, are indeed unmodified on eS7. We next tested, whether Otu2 plays a general role in rRNA maturation. Northern blotting and SYBR-Gold staining against mature 18S, pre-mature 20S rRNA as well as 25S rRNA of the LSU showed a general reduction of rRNA levels in otu2Δ and ubp3Δ strains, especially in the double knockout stain (Supplementary Fig. 8c). This result suggests that Otu2 indeed has a so far unknown function affecting ribosome assembly, which would be in agreement with the previous observation that its human homolog, OTUD6B, can be co-purified with late pre-40S particles from human cells[40].

### The N-terminal region of Otu2 provides specificity for 40S binding

Our structures show a dual mode to recognize a variety of 40S particles via the N-terminal domain and the C-terminal OTU domain: while all observed particles contain the N-terminal region of Otu2 comprising helices α1-α4, the OTU domain is only stably positioned in case eS7 is monoubiquitinated. We thus propose that 40S binding specificity is mainly contributed by the N-terminal domain and that eS7 mono-ubiquitination is not the major determinant to recruit Otu2 to the 40S. In agreement with this, we still find Otu2 bound to 40S in a strain where all four lysine residues close to the ubiquitination site of eS7 are mutated to arginine (eS7-4KR; Supplementary Fig. 9a, b).

Further insight into the role of Otu2 could be gained from our in vitro reconstituted Otu2-40S structure, which shows that presence of the Otu2-NTD alone on 40S would sterically clash with the 60S subunit in context of an 80S ribosome. Most intriguingly, α4 occupies the binding position of LSU protein eL19, that forms the functionally important intersubunit bridge eB12[70] (Fig. 4a, b). In 80S ribosomes eL19 connects the 60S with the 40S subunit by inserting with its long C-terminal α-helix into the major groove of expansion segment ES6e of the 40S. Helix α4 of Otu2 occupies a very similar position on ES6e, most likely involving basic amino acids including a patch of four consecutive lysines and arginines (Lys116-Arg119; BP1) (Fig. 4b and Supplementary Figs. 6c and 9c, d). We thus tested, if deletion of the C-terminal helix of eL19 would affect the specificity of Otu2 for 40S. Therefore, we generated yeast strains with 80S ribosomes lacking parts of or the entire C-terminal helix of eL19 (Supplementary Fig. 9c, d). Yet, Otu2 was still specifically found in the 40S but not in 80S fractions (Supplementary Fig. 9e), indicating that eL19-free eB12 bridge on 80S is not a major determinant for 40S specificity.

In addition to α4 of Otu2, that would clash with the 60S subunit, also the α1-α2 helical hairpin and α3 may contribute to 40S specificity (Fig. 4c, d). In context of the 80S ribosome, Otu2 binding is impaired by the tip of 25S rRNA H101 (ES41) of 60S subunit that is also involved in intersubunit bridge formation with eS8 (bridge eB11). Moreover, binding of the α1-α2 hairpin induces a small translational movement of the lower h44 towards the left foot to form the binding pocket. This would not be possible on an 80S ribosome, since h44 is involved in formation of the bridge eB13 with eL24 which stabilizes the overall 80S conformation.

To test if these overlapping binding sites may indeed provide specificity for Otu2 recruitment to 40S subunits, we generated

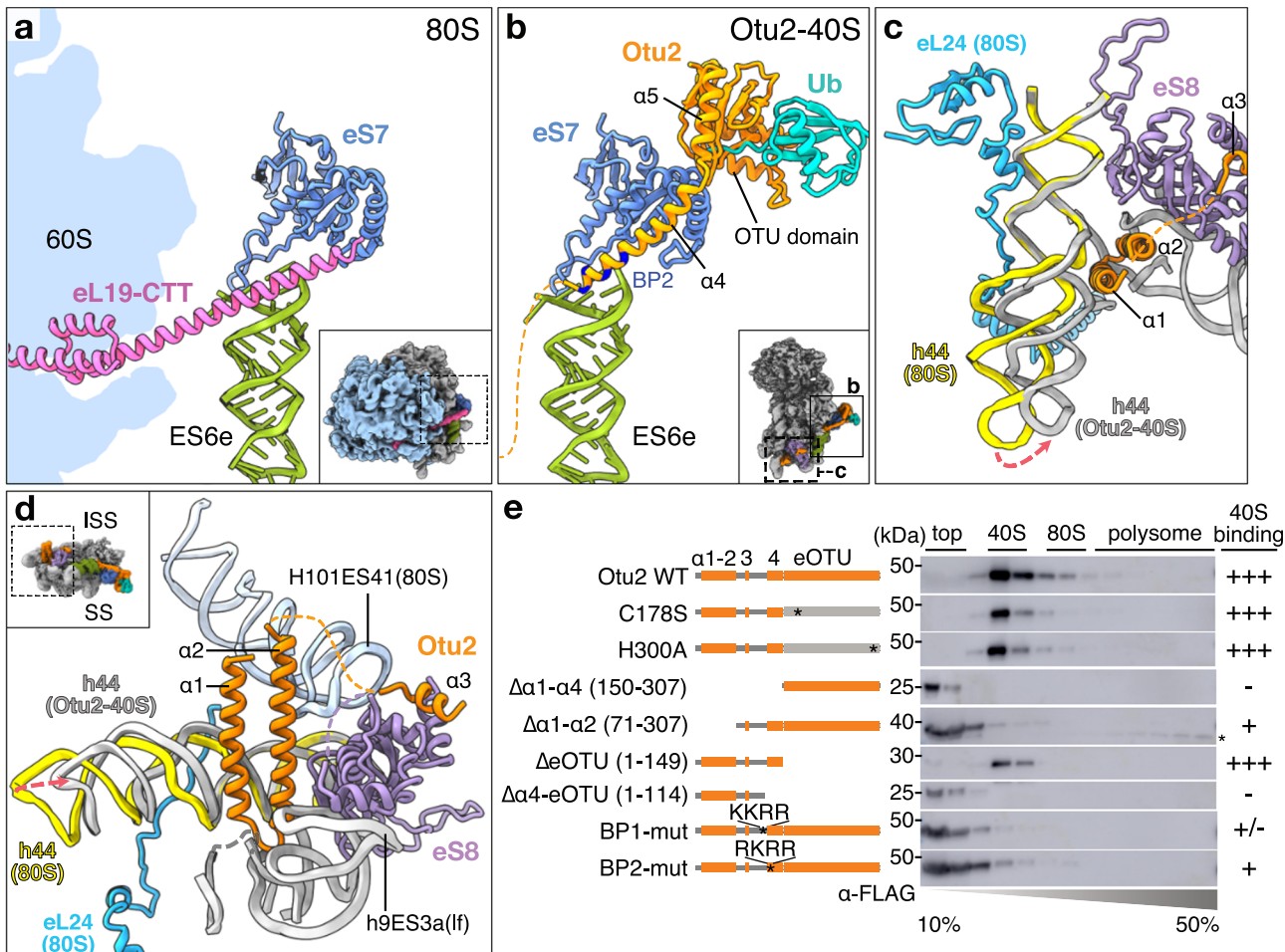

**Fig. 4 | Comparison between the 40S subunit in context of 80S and bound to Otu2. a** View on intersubunit bridge eB12 formed by the C-terminal tail (CTT) of 60S protein eL19 and the 40S ES6e as present in the non-rotated state 80S (PDB:6snt)[12]. **b** Same view as in **a** on the Otu2-bound 40S subunit. Instead of eL19, α4 of Otu2 binds to ES6e major groove; residues contained in basic patches (BP2) are indicated. Ub ubiquitin **c, d** views focusing on the Otu2 α1-α2 hairpin and α3 located close to intersubunit bridges eB13 (formed by 60S protein eL24 and 40S h44) and eB11 (formed by 60S H101/ES41 and eS8 of the 40S subunit) in 80S non-rotated state. The hairpin binds to h44 opposite of the eL24 binding site. When

bound to the hairpin, h44 moves towards the left foot (h9ES3a) thereby compacting the rRNA binding pocket for the hairpin. **e** Western blot analysis (α-FLAG) of fractions after sucrose density gradient centrifugation. Lysates were analyzed from an *otu2Δubp3Δ* yeast strain harboring a vector expressing wild-type (wt) Otu2 or indicated 3xFLAG-tagged Otu2 mutants. BP1-mut; K116A K117A R118A R119A, BP2-mut; R121A K123A R125A R129A. We obtained essentially the same results in at least three independent experiments. Original Blots are provided in the Source Data file.

truncation mutants of Otu2 and checked their association with 40S or 80S ribosomes in sucrose density gradients (Fig. 4e). While the vast majority of the wild type and catalytic triad mutant Otu2 (otu2-C178S and otu2-H300A) were found in the 40S fraction (Fig. 4e and Supplementary Fig. 2a), mutants lacking the entire N-terminus (Δα1-α4; 150–307) were not found in ribosomal fractions, indicating complete loss of 40S binding. Mutants without the α1-α2 hairpin (Δα1-α2; 71–307) but still containing α4 displayed residual but substantially reduced binding activity. We further found, that mutants lacking the extended OTU domain (ΔeOTU; 1–149, lacking α5) were still able to bind to 40S while shorter N-terminal fragments lacking α4 (Δα4-eOTU; 1–114) or carrying mutations in the basic patches close to or within α4 were not able to bind to 40S (BP1) or showed severely reduced binding (BP2) (Fig. 4e and Supplementary Fig. 6c).

Taken together, these data show, that the OTU domain alone has insufficient affinity for stable 40S binding, and that recruitment of Otu2 to the 40S is mediated by its N-terminus. Here, positioning of helix α4 at the ES6e binding site appears to be necessary but not sufficient for stable Otu2 binding to 40S, which requires the additional contribution of the N-terminal hairpin.

## Dynamics of Otu2 association with 40S

We next asked whether Otu2 could play an even more active role in the ribosome splitting process, either as a splitting factor similar to Dom34-ABCE1[43,45,71,72] or as anti-association factor resembling activity of ABCE1[41] on the 40S or of eIF6 on the 60S. We thus performed in vitro splitting and re-association assays using purified components (Supplementary Fig. 9f, g). Yet, unlike Dom34-ABCE1 or eIF6, even excess amounts of Otu2 did not promote splitting of 80S ribosomes into subunits (Supplementary Fig. 9f). Furthermore, Otu2 did not impair formation of 80S from 40S and 60S, as opposed to eIF6 (Supplementary Fig. 9g). This result suggests, that Otu2 has no active role in the 80S splitting process (e.g., by breaking the eB12 bridge) and binds the 40S only after subunit dissociation. Since in our native sample we observed Otu2 mainly bound to 40S complexes that carry other factors preventing 60S association, a redundant anti-association activity by Otu2 may not be required and it thereby would also not interfere with later coordinated subunit joining.

Taken together, our data show that stable recruitment of Otu2 to the ribosome depends on its N-terminal α-helices, with a particular contribution by the helical hairpin and helix α4. Since the interaction

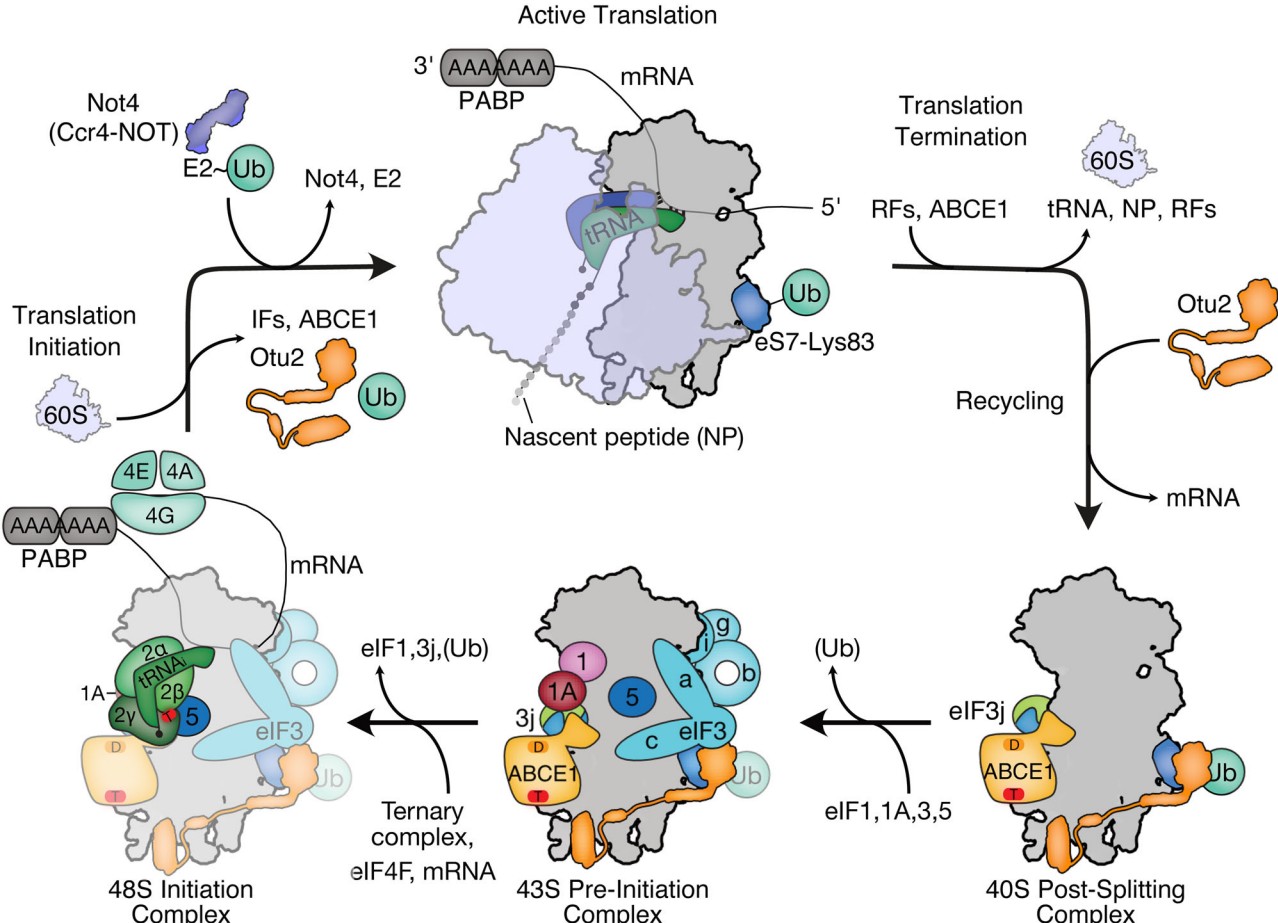

**Fig. 5 | Model for the eS7 ubiquitination-deubiquitination cycle during translation.** Shown is a simplified scheme of steps occurring between ribosome recycling and initiation: During active translation, eS7 is monoubiquitinated at Lys83. After termination, the 80S ribosome is dissociated into subunits and a 40S post-splitting complex is formed consisting of ribosome recycling factor ABCE1 in a hybrid conformation and eIF3j. The 43S pre-initiation complex forms by assembly of eIF1, eIF1A, eIF3, and eIF5, followed by binding of the ternary eIF2αβγ-tRNAi-GTP complex. mRNA loading leads to formation of the 48S initiation complex that scans the mRNA until the first AUG codon is recognized. This leads to positioning of the eIF5 N-terminal domain (eIF5-NTD) and dissociation of eIF1. Start codon recognition leads to dissociation of most initiation factors followed by subunit joining.

Otu2 engages the SSU as soon as the 80S ribosome is split into subunits and may remain bound to the 40S throughout all stages of initiation until subunit joining. Otu2 hereby plays a role in translational reset of recycled 40S subunits by deubiquitinating Lys83-monoubquitinated eS7. After subunit joining 80S ribosomes are marked as actively elongating ribosomes via eS7-ubiquitination by Not4 of the Ccr4-NOT complex. This marker can be further used as hub for eS7 polyubiquitination by Hel2 in case of aberrant translation and ribosome collision to cause No-Go mRNA decay. "D" and "T" in ABCE1 indicate presence of ATP and ADP in the hybrid state, that ABCE1 adopts in initiation complexes[50]. "T" in eIF2γ indicates bound GTP. Ub ubiquitin, PABP poly-A binding protein, NP nascent peptide, RFs release factors.

## Discussion

Mono-ubiquitination of eS7 has been shown to play an important role for gene expression regulation. Importantly, it enables the Ccr4-Not complex to be recruited to translating ribosomes and to function as a reader for codon optimality via Not5[28,31]. In vitro and also in living cells, Not4 was shown to add monoubiquitin to 80S ribosomes ([30]; this study) but not to 40S subunits[28]. Consistent with this previous study, our data show that Otu2 primarily acts during the 40S recycling and early (re-)initiation phase, as demonstrated indirectly by mass spectrometric analysis and directly by our cryo-EM shotgun approach. Here we find major populations of Otu2-bound eS7-monobiqitinated 40S in context of 43S pre-initiation complexes, the majority of which showed ABCE1 (Rli1) bound as described before[50]. Apart from ABCE1 we also find enrichment of the mRNA recycling factor Tma64 in our mass spectrometry, which is in agreement with the suggested role of Otu2 in mRNA recycling[30] (Fig. 5). Yet, apart from one small subclass that may

represent a Tma20-Tma22- or Tma64-bound mRNA recycling intermediate (Supplementary Fig. 3a), we did not find mRNA recycling factors in our cryo-EM analysis which may be explained by instability of the complexes and high dissociation activity. Contrary to the study of Takehara et al., we observed a 48S-IC subpopulation bound to Otu2. This functional state can be clearly assigned by the $P_{IN}$ state of tRNAi, that only occurs after start codon recognition and after the positioning of eIF5-NTD. Consistent with this observation, eIF5 was found coimmunoprecipitating in Otu2-pulldowns in the Takehara study. We, thus, conclude that Otu2 can remain associated with the 40S from early post-splitting stages throughout initiation and scanning for the AUG until start codon recognition (Fig. 5).

Of note, we find the small r-protein eL41 stably associated with all mature (post-recycling and initiating) 40S complexes, but not in the pre-40S. eL41 is incorporated into ribosomes during the late stage of cytoplasmic maturation of pre-60S[73] and was only found bound to the 40S in context with 80S ribosomes[74,75]. It is therefore likely that it is incorporated into the 40S after 60S joining. Thus, presence of eL41 in our mature 40S populations further indicates that those 40S underwent at least one translation cycle including a subunit recycling event.

In addition, we found Otu2 in context with late 40S biogenesis particles. Contrary to the initiating 40S complexes, eS7 was not monoubiquitinated in such particles, suggesting an additional role of Otu2 for 40S biogenesis, perhaps by preventing premature, unspecific eS7-ubiquitination. Yet, apart from a very general contribution of Otu2 to stability of rRNA (Supplementary Fig. 8c) we were not able to attribute more specific roles to Otu2 during 40S maturation.

Interestingly, we found Otu2 bound to a position on the 40S subunit, that has not yet been taken by any other observed 40S-binding factor. It thus is able to bind very generally to 40S subunits, irrespective of their particular functional state. Our structure rather suggests that Otu2 can engage the 40S subunit only as soon as 60S dissociation exposes the binding patches for the N-terminal domain of Otu2. Here, the major groove of ES6e of the 18S rRNA plays an important role which in the 80S ribosome is occupied by eL19 of the 60S subunit, but on the free 40S subunit can accommodate helix α4 of Otu2. This interaction appeared necessary but not sufficient for efficient Otu2 binding since additional binding specificity is added by the N-terminal α1-α2 helical hairpin. This is of particular interest since it is not strictly required for 40S binding but induces a conformational reorganization of the rRNA h44, thereby potentially contributing to allosteric communication with factors bound at or close to the decoding center of h44 in pre-40S, recycling 40S and all 43S/48S particles.

Taken together, our findings make a contribution to the molecular understanding of the dynamic Otu2 association with the small ribosomal subunits: Otu2 can associate with multiple states of ribosomal particles, either pre-40S or any 40S after ribosome splitting into subunits until the stage of start-codon recognition. Specificity for these 40S is established via the Otu2 NTD that binds to a site not occupied by other factors independent on the eS7-ubiquitination state. While our structure shows that the Otu2-NTD binding is mutually exclusive with 60S binding, our experimental data suggest that the presence of 60S subunits may be sufficient to could trigger Otu2 dissociation from 40S subunits. We thus speculate that Otu2 is a constitutive 40S binding protein, maybe serving as an additional checkpoint for successful initiation or 40S biogenesis, that only dissociates after initiation or 40S maturation reach completion.

Only 21 deubiquitinating enzymes exist in yeast in total (two of the OTU subfamily) which mostly cover a broader substrate spectrum[76]. In contrast, we observed that Otu2 was inactive for any types of diubiquitin cleavage and rather exerts a remarkable specificity for 40S monoubiquitinated eS7 at Lys83, which we can now explain in molecular detail.

Overall, our data provide the structural basis for the general translational reset step by deubiquitination of eS7 during recycling/(re)initiation. Yet, the precise interplay of Otu2 with the recycling/initiating 40S and the trigger for dissociation will be subject of future studies.

## Methods

### Yeast strains and plasmids

All yeast strains used in this study were derivatives of the budding yeast (*Saccharomyces cerevisiae*) W303-1a parental strain (see also Supplementary Table 3 and 4 in Supplementary Information). For label-free quantification mass spectrometric analysis, yeast strains expressing FLAG-TEV-proteinA (FTpA)-tagged Otu2 and Ubp3 were generated by established yeast techniques[77]. Gene deletion yeast strains were constructed by homologous recombination using amplified selection marker DNA fragments[78]. The coding sequence and the promoter of *OTU2* were cloned from W303-1a genomic DNA into the p*415-3xFLAG-CYC1t* vector. The Otu2 point mutants were generated by site-directed mutagenesis. N- or C-terminal Otu2 deletion mutants were created by PCR using specific oligonucleotides combinations. Otu2 expression plasmids were transformed into *otu2Δ* or *otu2Δubp3Δ*

yeast strains using lithium acetate, then grown and selected on SDC -Leu medium plates. To obtain wild-type (wt) Otu2 and otu2-C178S, *OTU2* and *otu2-C178S* were cloned into a pGEX6P-1-His$_6$-GST-3C expression vector. For recombinant Not4, Ubc4 and Uba1, pGEX6P-2-*NOT4-FLAG*, pGEX6P-1-*UBC4* vectors were used for expression in *E. coli* and the p416*GPDp-UBA1-FLAG* vector was used for expression in a yeast *leu1Δ* strain. To generate eS7A-HA-tagged 80S ribosomes, *otu2Δrps7aΔrps7bΔ* yeast strains harboring a eS7A-HA expressing plasmid[4] were used.

For the native Rio2 pullout, an eS7 shuffle strain harboring an eS7A-HA expression plasmid was genomically tagged to express His-TEV-proteinA (HTpA)-tagged Rio2. To prepare the eL19-shuffle strain, we first knocked out the *RPL19A* gene by homologous recombination, second transformed the *rpl19aΔ* strain with p416-RPL19-CYC1t plasmid for sufficient eL19 protein expression, third deleted the *RPL19B* gene which yields the eL19-shuffle strain. To exchange the eL19 plasmid from p416-*RPL19-CYC1t* to p415-*RPL19-CYCt* (full-length or truncated mutants), we transformed the eL19-shuffle strain with p415-*RPL19-CYCt* plasmids, grew them on SDC-Leu agar plates followed by the strain selection on a SDC + 5-FOA (5-fluoroorotic acid) plate. The coding sequence and the promoter of *RPL19* and its mutants were cloned from W303-1a genomic DNA into the p*416/415* vector. Because of high similarity between *RPL19A* and *RPL19B* genes, the *RPL19* clone used in this study has partly chimeric sequence at DNA level but the derived protein sequence is the same as of the eL19A protein.

### Preparation of native Otu2-bound 40S complexes

Yeast cells expressing C-terminally 3xFLAG-tagged otu2-C178S were cultivated in 2 l SDC -Leu liquid medium, grown at 30 °C until an OD$_{600}$ of 0,8, harvested by centrifugation and quickly frozen in liquid nitrogen. The frozen yeast cell pellet was ground in liquid nitrogen using a mortar and a pestle. Resulting yeast cell powder was resuspended in ice-chilled lysis buffer-DG (20 mM HEPES-KOH pH 7.5, 100 mM KOAc, 2 mM Mg(OAc)$_2$, 1 mM dithiothreitol (DTT), 1 mM phenylmethylsulfonyl fluoride (PMSF), 1 tablet per 10 mL of cOmplete tablets EDTA-free (Roche)), spun down in a table top centrifuge to remove the rough cell debris, and further centrifuged at 40,000×*g* in a SS-34 rotor to remove debris. The yeast lysate was loaded on a 10–50% sucrose density gradient in SW32 tubes, and ultracentrifuged at 35,000×*g* for 16 h. After fractionation with continuous detection of A$_{260}$ the 40S fraction was collected, concentrated using an Amicon-100K and the buffer was changed to buffer EM (20 mM HEPES-KOH pH 7.6, 50 mM KOAc, 5 mM Mg(OAc)$_2$, 50 mM sucrose, 1 mM DTT, 0.05% Nikkol). Cryo-EM grids were prepared using concentrated 40S fraction samples. The final concentration of the sample was 9.2 A$_{260}$/ml.

### In vitro reconstitution of Otu2-40S complexes

For the preparation of ubiquitinated 40S ribosomal subunits, the yeast eS7A-HA-shuffled strain was cultivated in 2 l of YPD and harvested at an OD$_{600}$ of 1.5. Yeast crude ribosomes were purified from the yeast lysate by pelleting through a sucrose cushion. The crude ribosomal pellet was resuspended in resuspension buffer (20 mM HEPES-KOH pH 7.5, 100 mM KOAc, 5 mM Mg(OAc)$_2$, 1 mM DTT, 1 mM PMSF) and treated with micrococcal S7 nuclease (Thermo Fisher Scientific, cat# EN0181) at a final concentration 40 U/mL in presence of 0.8 mM CaCl$_2$ at 25 °C for 15 min. The reaction was stopped by adding 2 mM EGTA. To purify 80S ribosomes, the S7-treated ribosomes were separated using 10–50% sucrose density gradient centrifugation with a SW32 Ti rotor at 125,000×*g*, 4 °C for 3 h. The 80S fraction was further centrifuged in a TLA110 rotor at 335,000×*g*, 4 °C for 45 min. The resulting ribosomal pellet was resuspended in 250 μL resuspension buffer (-65 A$_{260}$/ml).

1.25 pmol of purified 80S ribosomes were subjected to test the in vitro ubiquitination reaction with 50 μM ubiquitin, 100 nM Uba1, 300 nM Ubc4, 180 nM Not4 and energy regenerating source (1 mM ATP, 10 mM creatine phosphate, 20 μg/ml creatine kinase) in buffer-UB

(20 mM HEPES-KOH pH 7.5, 100 mM KOAc, 5 mM MgCl$_2$, 1 mM DTT) in 25 μl reaction volume at 26 °C for 60 min. The reaction was analyzed by NuPAGE and Western blotting. To prepare Ub-40S for in vitro reconstitution, 50 pmol of purified 80S ribosomes were used for the ubiquitination reaction as described above with respectively higher concentration of Ubc4 (5 μM) and Not4 (1 μM), incubated at 28 °C for 90 min, then the tubes were placed on ice to stop the reaction.

To release peptides and tRNAs from ribosomes, a high salt/puromycin treatment was performed and resulting ribosomes were loaded onto a 5–20% sucrose density gradient (5–20 % sucrose, 20 mM HEPES-KOH pH 7.5, 500 mM KCl, 2 mM MgCl$_2$, 2 mM DTT) to split the ubiquitinated ribosomes into subunits. The buffer of the 40S fraction was exchanged in an Amicon Ultra-100K to storage buffer (20 mM HEPES-KOH pH 7.5, 100 mM KOAc, 2.5 mM Mg(OAc)$_2$, 2 mM DTT). Ubiquitination of eS7 was confirmed by western blotting. The final concentration of Ub-40S was 7.6 A$_{260}$/ml in 100 μl.

### In vitro deubiqitination assay
Ub-40S was used for an in vitro deubiquitination assay, in which 1 μl Ub-40S with 1.5 μM Otu2 in storage buffer was incubated at 30 °C, for 60 min. Presence of Ub-eS7A-HA was analyzed by western blotting using α-HA antibodies. For reconstitution of the otu2-C178S-Ub-40S complex used for cryo-EM analysis, 12 pmol Ub-40S and 5-fold molar excess of otu2-C178S were mixed and incubated at 30 °C for 10 min, placed on ice and cryo-EM grids were prepared.

For the di-ubiquitin cleavage assay, we used the di-ubiquitin Explorer Kit (Cat#J2000, UBPBio). 3 μM di-ubiquitin and 0.5 μM Otu2 were mixed in storage buffer containing 5 mM DTT. Samples were analyzed by Nu-PAGE and gels were stained by Der Blaue Jonas (Cat#GRP1, GRP).

### Western blotting and antibodies
After NuPAGE or SDS-PAGE, western blotting was performed with the semi-dry method. The membrane was then incubated with 5% skim milk in phosphate buffered saline containing 0.1% (w/v) Tween-20 (5% milk in PBS-T) for 1 h, and further incubated with horseradish peroxidase (HRP)-conjugated specific antibodies in 5% milk in PBS-T. After three times washing the membrane with PBS-T, proteins were detected with the AI-600 image (GE Healthcare) using SuperSignal West Dura Extended Duration Substrate (Cat# 37071, Thermo). Antibodies targeting FLAG (1:5,000; Monoclonal ANTI-FLAG M2-peroxidase, Cat# A8592, Sigma), HA (1:5,000; anti-HA-peroxidase, high Affinity, 3F10, Cat# 1201381900, Roche) and His (1:1,000; anti-His tag antibody (mouse monoclonal, Cat#G020, abm) were used.

### Electron microscopy and image processing
Copper grids with holey carbon support film (R3/3, Quantifoil) and a 2 nm pre-coated continuous carbon layer on top were glow discharged at 2.2 × 10$^{-1}$ mbar for 20 s. 3.5 μl of sample was applied to the grid in a Vitrobot Mark IV (FEI Company), blotted for 2 s after 45 s of incubation at 4 °C and flash frozen in liquid ethane. Data were collected on a Titan Krios at 300 keV using EPU software (version 2.12.1). For the in vitro reconstituted Otu2-Ub-40S complex sample, 5726 movies were collected with a pixel size of 1.059 Å/pixel and within a defocus range of 0.5–4.0 μm using a K2 Summit direct electron detector under low-dose conditions with a total dose of 46 e$^-$/Å$^2$. For the native Otu2-bound complexes sample, 6441 movies were collected within a defocus range of 0.5–3.0 μm with a total dose of 46.4 e$^-$/Å$^2$. Gain-corrected movie frames were motion corrected and summed with MotionCor2[79] and Contrast Transfer Function (CTF) parameters were determined with CTFFIND4[80]. 540,175 (in vitro) and 656,489 (native) particle images were picked by Gautomatch with 2D projections of 40S particle images (https://www2.mrc-lmb.cam.ac.uk/?s=gautomatch). After exclusion of ice and bad particles by 2D classification in Relion 3.0 (in vitro) and 3.1 (native)[53], 316,588

(in vitro) and 535,873 (native) 40S/43S/48S particle images were obtained. Good particles were selected, 3D refined, and classified as shown in Supplementary Figs. 3a and 4f.

For the in vitro reconstituted sample, 40S particles were first classified for presence of extra density for the Otu2 N-terminal hairpin on the 40S foot structure. This was done using focused classification with a soft spherical mask for this region. In all, 10% of particles (31,646) showed clear extra density not only for the N-terminal hairpin but also for the OTU domain of Otu2 located on eS7. This population was subjected to multi-body refinement on the 40S head and 40S body part including Otu2. The final maps were post-processed and local resolution was estimated with Relion 3.1. All maps were shown low-pass filtered according to local resolution estimation with b-factor of 20 applied in Relion 3.1.

For the native complexes sample, three branches of image processing were performed. In the first branch (upper right panel in Supplementary Fig. 3a), 3D classification was performed without mask in Relion 3.1 (3D classification I). 7.5% of particles (40,334) fell into a class representing well-resolved 43S PIC, 7.2% (38,748) a partial 48S IC and 3.4% a pre-40S. Notably, 12.6% particles showed 40S with weak extra density in a position where the MCR-1/DENR and eIF2D, homologs of yeast Tma20-22 or Tma64, were previously identified[81,82]. We thus designated this class as "Tma-like". Other classes also showing (weak) density for 43S PIC/48S IC components were not processed further.

For the 48S IC class, a focused classification (3D classification I-a) was performed on the extra density for the Otu2-N terminal hairpin between the 40S foot structure, and 12,059 particles were classified as Otu2-bound Ub-48S IC. For the "Tma-like" class, further focused classification (3D classification I-b) was performed and 22,640 particles contained stronger "Tma-like" density. Because of flexibility and low-resolved extra density, we were neither able to sort further nor distinguish between Tma20/22 and Tma64.

In the middle left branch, focused classification jobs in Relion 3.1 were first performed on the extra density for Otu2-N terminal hairpin (3D Classification II) and the OTU domain at eS7 (3D Classification III). Here, 19.7% of particles (103,590) were enriched in Otu2-N hairpin bound to the 40S foot structure and 18.7% of particles (100,907) were enriched in Otu2 C-terminal OTU domain with ubiquitinated eS7. To obtain higher resolution of Otu2-N and -C with Ub-eS7, principal components analysis (PCA)-based 3D variability analyses were performed in CryoSPARC v3.3.2[51,52] (3D var in CS, lower right) with soft masks covering the respective Otu2 densities, and 36,086 particles were enriched in Otu2-N and 31,008 particles in Otu2-C with Ub-eS7. After local refinement, post-processing and local resolution filtering with a b-factor of −64.1 and −66, respectively, the final average resolution of the maps at 0.143 FSC threshold was estimated to 2.9 Å and 3.0 Å, respectively. Corresponding local resolution ranges were -2.3–5.0 Å for Otu2-N, 4.0–7.0 Å for Otu2-C, and ubiquitin and 3.0–4.0 Å for eS7.

Following 3D Classification II, 3D Classification II-a was performed without application of a mask (lower left branch), and 18.2% of input particles (18,826) were sorted as well-resolved Otu2-bound Ub-43S PIC and 4.7% (4,908) as Otu2-bound pre-40S. The Otu2-bound Ub-43S class was further sub-classified with a soft mask covering the region of ABCE1 resulting in 70.8% of Otu2-bound 43S-PIC particles (13,427) in the ABCE1-bound state. Finally, all three structures were subjected to 3D refinement, CTF refinement and/or non-uniform refinement in CryoSPARC v3.3.2. Obtained maps were filtered according to local resolution using a b-factor −46 for 43S PIC, −47 for 48S IC and −40 for pre-40S and the final average resolution was estimated to 3.0 Å (43S PIC), 3.3 Å (48S IC) and 3.8 Å (pre-40S), respectively.

To obtain figures, molecular models and cryo-EM densities were displayed in ChimeraX (v.1.3)[83]

## Otu2 (150–307) purification for crystallization

A pGEX plasmid containing the His$_6$-GST-HRV 3C-tag and the gene for *S. c.* Otu2 C178S was modified by iPCR[84] to code for the Otu2 globular domain (residues 150–307). The protein was expressed in a *E. coli* Rosetta2 DE3 strain. Cells were grown in LB media supplemented with antibiotics at 37 °C until an OD$_{600}$ of 0.4–0.6 was reached. The temperature was then lowered to 18 °C and 0.5 mM IPTG was used for induction overnight. Cells were pelleted, resuspended in lysis buffer (50 mM sodium phosphate buffer pH 8.0, 500 mM NaCl, 10% (v/v) glycerol and 20 mM imidazole pH 8.0) and lysed using a cell homogenizer. Soluble proteins were separated from the cell debris by centrifugation (20 min, 40,000×$g$). The supernatant was incubated with Ni-NTA resin for 20 min at 4 °C, washed extensively with lysis buffer and 4 column volumes of lysis buffer containing 40 mM imidazole pH 8.0. The protein was then eluted in lysis buffer containing 300 mM imidazole and dialysed overnight at 4 °C against cleavage buffer (20 mM Tris pH 8.0, 150 mM NaCl, 1 mM DTT) and HRV 3 C protease was added. To separate the His$_6$-GST tag from Otu2, the sample was again incubated with Ni-NTA and the unbound fraction containing only the cleaved Otu2 globular domain was collected and concentrated to 3 mL. The sample was then loaded on HiLoad 16/600 Superdex 75 pg size-exclusion column. Peak fractions were collected, concentrated to around 20 mg/mL and flash-frozen in liquid nitrogen.

Plate-like crystals of the Otu2 C178S globular domain were obtained at 20 °C in hanging drops by mixing 1 µL of protein (22 mg/mL) and 1 µL of crystallization solution (0.2 M KSCN, 0.1 M BisTris propane pH 6.5, 20% (w/v) PEG 3350 and 20% (v/v) glycerol). Crystals were flash frozen in liquid nitrogen and crystallographic data were collected at SLS (Swiss Light Source) synchrotron. X-ray data were processed with XDS software[85] and the structure was solved with the molecular replacement method using the program Phaser (part of the PHENIX suite, version 1.14)[86,87] and the structure of the OTUD3 OTU domain (PDB ID 4BOU)[58] as the search model. Automated model building was done with the AutoBuild software in PHENIX[88] and manually in COOT (version 0.8.9)[89]. Restrained refinement was performed with phenix.refine of the PHENIX package[87]. Accuracy of the model was assessed using MolProbity of the PHENIX package. Final statistics can be found in Supplementary Table 2.

## Model building

The model for the Otu2-bound Ub-43S-PIC and Ub-48S-IC was based on previous structures of either the *S.c.* 43S-PIC (PDB:6ZCE) or the *S.c.* 48S-IC (PDB:6ZU9)[50]. These structures were first rigid-body fitted into the density and manually adjusted in COOT (version 0.8.9)[89] in regions where the resolution allowed fitting of side chains. To match the conformation of eIF2, PDB 6FYY[65] was used as template for rigid-body fitting.

For the Otu2-bound pre-40S, the *S.c.* pre-40S from PDB 6FAI[67] served as a template. The model for Dim1 was extracted from PDB 6RBD[90].

A model for eS7-mono-Ub-bound Otu2 was obtained as follows: We first fitted our X-ray structure of the extended OTU domain (150–307) into the respective density close to eS7. Here, we could also unambiguously identify ubiquitin based on secondary structure as demonstrated in Supplementary Figs. 3c, 5k, and 7c, e). For the Otu2 loop β3-β4 (Ser294-Gly298), that was not resolved in the X-ray structure (Supplementary Fig. 6f), the AF2 model served as a template for manual adjustment to fit into the cryo-EM density (Supplementary Fig. 7d).

The Otu2 N-terminus was fitted into the cryo-EM map using a model generated by AlphaFold2 (AF2)[55,91]. The predicted N-terminal hairpin (α1-α2; res 14–69) as well as α3 (84–94) and the long α4 fitted with only minor adjustments into the respective densities. The part of h44 close to the N-terminal hairpin of Otu2 was manually adjusted to fit the density using COOT.

Because the resolution did not allow unambiguous placement of sidechains of Otu2, we refrained from including them in the final model, while preserving the residue register. Validation of the final model was done using PHENIX[87].

For the reconstituted 40S-Ub-Otu2 structure the *S.c.* 40S subunit from PDB-6TB3[31] served as a template. Since the 3D reconstruction was focused on the 40S body, the resolution of the head region of the 40S was considerably lower. We therefore removed proteins and rRNA belonging to the head from our model. Due to low resolution in the platform region, we also removed proteins and rRNA belonging to the 40S platform. None of the removed parts relevant in the context of Otu2 binding.

## Polyribosome profile analyses

Yeast cells (*otu2Δ*, *otu2Δubp3Δ*, *eS7-WT*, *eS7-4KR,* or eL19-shuffle strains) expressing wt or mutant Otu2-3xFLAG from plasmids or expressing Otu2-FTpA or Otu2-3xFLAG from the genome were cultivated in 200 ml SDC -Leu or YPD at 30 °C until an OD$_{600}$ of 0.6, harvested by centrifugation and flash-frozen in liquid nitrogen. Ground yeast cell powder – dissolved in buffer-DG containing 0.1 mg/ml cycloheximide - was centrifuged in a table-top centrifuge at 4 °C to remove cell debris. The obtained lysate was then loaded on a 10–50% sucrose gradient (10 mM Tris-OAc pH 7.5, 70 mM NH$_4$OAc, 4 mM Mg(OAc)$_2$) in SW40 tubes and centrifuged at 200,000×$g$, 4 °C for 2.5 h. Samples were analyzed using a continuous UV detector at 260 nm connected to piston fractionator (biocomp). Each fraction was mixed well with trichloroacetic acid at final concentration 10%, placed on ice for 15 min, centrifuged at 20,000×g for 15 min, then the supernatant was completely discarded. The protein pellet was dissolved in 1x basic sample buffer (120 mM Tris, 3.5% SDS, 14% glycerol, 8 mM EDTA, 120 mM DTT, 0.01% bromophenol Blue (BPB)), and incubated at 99 °C (Otu2) or 88 °C (ubiquitinated eS7A) for 10 min. Soluble fractions were analyzed by SDS-PAGE and Western blotting using α-FLAG and α-HA antibodies.

## Spot assays

Yeast cells grown in YPD for 24 h were diluted at OD$_{600}$ of 0.3, and a series of 10x dilutions was prepared for each sample. 2 µL each of dilution was spotted on YPD plates in absence or presence of translation elongation inhibitors (10 µg/ml anisomycin or 50 ng/ml cycloheximide), and the plates were incubated in 30 °C or 37 °C for 2 days.

## Affinity purification for mass spectrometry analysis

W303-1a as a non-tag control, Otu2-FTpA, and Ubp3-FTpA yeast cells were grown in 4–6 L YPD until a OD$_{600}$ of 1.5 and three independent cultures were prepared as biological replicates for both Otu2 and Ubp3. Cells were harvested by centrifugation, frozen in liquid nitrogen and lysed using a mortar and a pestle. The resulting powder was dissolved in lysis buffer 7.5 (20 mM HEPES-KOH pH 7.6, 100 mM KCl, 10 mM MgCl$_2$, 1 mM DTT, 0.5 mM PMSF, 0.01% NP-40, 1 pill/13 ml of cOmplete tablets EDTA-free). After centrifugation at 40,000× $g$, 4 °C for 25 min, the lysate was incubated with rabbit IgG-conjugated Dynabeads M-270 epoxy (invitrogen) for 1 h at 4 °C on a nutator. The beads were then washed seven times with lysis buffer and treated with S7 nuclease (40 U/ml) at 25 °C for 15 min in presence of 0.8 mM CaCl$_2$. The S7 reaction was stopped by adding 2 mM EGTA, and the eluate was stored as the "S7-wash" sample. After washing the beads once by lysis buffer, the beads were incubated with His-TEV protease at 4 °C for 1.5 h on a rotator. The eluate was collected as "TEV" sample. Both samples were subjected to TCA precipitation, and the protein pellet was recovered in 1x sample buffer pH 6.8 (50 mM Tris-HCl pH 6.8, 2% SDS, 10% glycerol, 0.01% BPB, 25 mM DTT). After denaturation by incubation at 95 °C for 10 min and centrifugation at 20,000×$g$ for 10 min, soluble fractions were analyzed by Nu-PAGE gel and stained by SimplyBlue SafeStain (invitrogen).

## Affinity purification of Rio2-bound complexes

Yeast Rio2-HTpA eS7-shuffle (eS7A-HA) strains harboring empty vector, Otu2 wt or C178S plasmids were grown in 2 l SDC -Leu liquid media at 30 °C until $OD_{600}$ of 0.8 and harvested by centrifugation. After cell grinding by a mortar and a pestle in liquid nitrogen, cell powder was resuspended in lysis buffer 7.5 (20 mM HEPES-KOH pH 7.6, 100 mM KCl, 10 mM $MgCl_2$, 1 mM DTT, 0.5 mM PMSF, 0.01% NP-40, 1 pill/20 ml of cOmplete tablets EDTA-free). After centrifugation at $40,000 \times g$, 4 °C for 25 min, the lysate was incubated with rabbit IgG-conjugated Dynabeads M-270 epoxy (Cat#14301, invitrogen) for 1 h at 4 °C on a nutator. The beads were washed seven times with lysis buffer, and incubated with His-TEV protease at 4 °C for 2 h on a rotator. The TEV elution samples were analyzed by Nu-PAGE, Der Blaue Jonas (Cat#GRP1, GRP) staining and western blotting using antibodies for epitope tags.

## Electrophoresis and northern blotting for ribosomal RNAs

Total RNA samples were extracted from 10 ml scale cultured yeast cells by using water-saturated RNA hot phenol and phenol-chloroform-isoamylalcohol (25:24:1). 5 µL of RNA (approximately 4 µg) were mixed with RNA loading buffer (30 mM Tricine, 30 mM Triethanolamine, 0.5 M formaldehyde, 5% (v/v) glycerol, 1 mM EDTA, 0.005% (w/v) Xylene cyanole, 0.005% (w/v) bromophenol blue in deionized formamide) and incubated at 65 °C for 5 min, then placed on ice for 5 min. rRNAs were separated on a 1% agarose gel with 1x TT buffer (30 mM Tricine, 30 mM Triethanolamine, pH 7.9 in 50x stock) by electrophoresis at 110 V for 120 min. For SYBR Gold staining the gel were soaked in 1/10,000 SYBR Gold-containing 1x TT buffer. For Northern blotting, RNAs in gels were capillary transferred onto a Hybond-N+ membrane (cytiva #RPN303B) using 20x SSC (3 M NaCl, 300 mM Sodium citrate). Hybridization was performed at 52 °C using DIG-labeled ITS1 probe and *SCR1* probe in 5 ml DIG easy hyb granules (Roche # 11796895001) and incubated at 50 °C in a hybridization oven for 20 h. The membrane was washed once for 15 min with 2x SSC 0.1% SDS, twice for 15 min each with 0.1x SSC 0.1% SDS in the oven, then incubated in 1x blocking reagent (Roche #11096176001) for 30 min and with 1/10,000 anti-digoxigenin-AP (Roche #11093274910) for 1 h at room temperature. After washing the membrane three times with wash buffer (100 mM maleic acid, 150 mM NaCl, 0.3% Tween-20, pH 7.5) and once with pre-detection buffer (0.1 M Tris-HCl pH 9.5, 0.1 M NaCl), RNA was detected via chemiluminescence using CDP-*star* reagent (Roche #11759051001) on AI-600 mini (GE healthcare) using incremental mode with 5 min intervals.

## In vitro splitting assay and reassociation assay

Splitting and reassociation assays were set up in principle as described in Wells et al. and Heuer et al. [41,92]:Puromycin-treated 80S ribosomes were prepared as described above for the in vitro reconstitution of Otu2-40S complex. 5 pmol 80S ribosomes were mixed with 5-fold molar excess of eIF6, eIF6 + Otu2 and eIF6 + Dom34-Hbs1-ABCE1 with 1 mM ATP and 1 mM GTP in Splitting buffer (20 mM HEPES-KOH, 100 mM KOAc, 7.5 mM Mg(OAc)$_2$). After the splitting reaction, performed at 25 °C for 10 min first and then on ice for 10 min, samples were analyzed by a 10–50% sucrose density gradient (described in polysome analysis section) centrifugation at $200,000 \times g$ for 4 h. Continuous $A_{260}$ detection data was plotted and analyzed.

For reassociation assays, 5 pmol of each yeast 40S and 60S ribosomal subunits were mixed with 5-fold molar excess of Otu2 or eIF6 in Reassociation buffer (20 mM HEPES-KOH, 100 mM KOAc, 12.5 mM Mg(OAc)$_2$) and incubated for 10 min at 25 °C, then placed on ice for 10 min. Samples were analyzed by the same method as described above.

## Sample preparation for mass spectrometry

Affinity purified samples ($n = 3$ per group prepared individually) were transferred to a Nu-PAGE gel and run for 6 min at 200 V until the gel pockets were empty. Gels were stained for 60 min using SimplyBlue Safestain and protein-containing areas were excised. To reduce proteins, gel bands were treated with 45 mM dithioerythritol in 50 mM $NH_4HCO_3$ for 30 min at 55 °C. Free sulfhydryl groups were carbamidomethylated by $2 \times 15$ min incubation in a solution of 100 mM iodoacetamide in 50 mM $NH_4HCO_3$ at room temperature. Prior to digestion, gel slices were minced. Digestion was performed for 8 h at 37 °C using 70 ng modified porcine trypsin (Promega, Fitchburg, WI, USA). Tryptic peptides were extracted using 70% acetonitrile and dried using a SpeedVac vacuum concentrator.

## Mass spectrometry analysis

Peptides were analyzed with an Ultimate 3000 nano-liquid chromatography system (Thermo Fisher Scientific) online-coupled to a Q Exactive HF-X mass spectrometer (Thermo Fisher Scientific). Peptides were diluted in 15 µl 0.1% formic acid and injected on an Acclaim PepMap 100 trap column (nanoViper C18, 2 cm length, 100 µM ID, Thermo Scientific). Separation was performed with an analytical EasySpray column (PepMap RSLC C18, 50 cm length, 75 µm ID, Thermo Fisher Scientific) at a flow rate of 250 nl/min. 0.1% formic acid was used as solvent A and 0.1% formic acid in acetonitrile was used as solvent B. As chromatography method a 30 min gradient from 3% to 25% solvent B followed by a 5 min gradient from 25% to 40% B was used. Data dependent mass spectrometry was performed using cycles of one full MS scan (350 to 1600 m/z) at 60k resolution and up to 12 MS/MS scans at 15k resolution. Acquired MS spectra were analyzed with MaxQuant (1.6.1.0)[93] and the *Saccharomyces cerevisiae* subset of the UniProt database (downloaded 2nd of December 2020). For the MaxQuant Andromeda search, the following parameters were used: Enzyme: Trypsin/P; missed cleavages ≤2; 4.5 ppm mass tolerance precursor for main search; 20 ppm mass tolerance MS/MS; minimum peptide length 7; carbamidomethylation of cysteine as fixed modification and oxidized methionine as well as acetyl (Protein N-terminus) as variable modification. FDRs at the peptide and protein level were set to 1%. LFQ values were used for label-free quantification. Only proteins identified with 2 unique/razor peptides were included for quantification. All replicates ($n = 3$ per group) were measured twice leading to six LC-MS/MS runs per group. The mass spectrometry proteomics data have been deposited to the ProteomeXchange Consortium (http://proteomecentral.proteomexchange.org) via the PRIDE partner repository[94].

## Relative protein quantification and statistics

Data analysis and statistics was done using Perseus (1.5.3.2)[93]. To handle missing values, the imputation feature of Perseus was used. Missing values were imputed from a normal distribution (width, 0.3; downshift, 1.8). For statistical evaluation, a two-sided Student's $t$ test including a permutation-based FDR correction was performed. Significant hits (FDR < 0.05) with log2-fold changes < −0.6 and >0.6 were regarded as relevant. The list of differently abundant proteins and corresponding quantitative values can be found in Source Data.

## Reporting summary

Further information on research design is available in the Nature Portfolio Reporting Summary linked to this article.

# Data availability

The mass spectrometry proteomics data have been deposited to the ProteomeXchange Consortium (http://proteomecentral. proteomexchange.org) via the PRIDE partner repository with the dataset identifier PXD041573. The cryo-EM structural data generated in this study have been deposited in the Protein Data Bank and the Electron Microscopy Data Bank under EMDB accession codes EMD-16470 and EMD-16471 for the in vitro reconstituted Otu2-Ub-40S body and head, EMD-16525 for the Otu2-Ub-43S-PIC, EMD-16533 for the

Otu2-Ub-48S-PIC, EMD-16541 for the Otu2-pre-40S, EMD-16542 for Otu2-N and EMD-16548 for Otu2-C and Ub-eS7), and PDB accession codes 8C83 for the Otu2-Ub-40S body, 8CAH for the Otu2-Ub-43S-PIC, 8CAS for the Otu2-Ub-48S-PIC and 8CBJ for the Otu2-pre-40S. The crystal structure of the extended OTU domain of Otu2 has been deposited in the Protein Data Bank under accession code 7PL7. Source data are provided with this paper.

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

## Acknowledgements

We thank M. Kösters, C. Ungewickell and S. Rieder for excellent technical assistance; P. Tesina for cryo-grid preparation; H. Kratzat for (pre-)initiation complex model illustration; M. Thoms for providing yeast strains and plasmids; Y. Takehara and S. Murata for communication and discussion; the crystallization facility of the Max-Planck-Institut (Conti Department) and Dr. Jérôme Basquin for the help with the X-ray data collection. This study was supported by JSPS Overseas Research Fellowship to K.I., a Ph.D. fellowship by Boehringer Ingelheim Fonds to R.Bu., grants by the DFG to R.B. [SFB/TRR-174, BE1814/15-1, BE1814/1-1], by the European Research Council grant Human-Ribogenesis (No. 885711) to R.B., by AMED Grant Number JP 19gm1110010 and JP223fa627001 to T.I., by MEXT/JSPS KAKENHI Grant Numbers JP18H03977, JP19H05281 to T.I., 21H00267, 21H05710, and 22H02606 to Y.M., by JST PREST Grant Number JPMJPR21EE to Y.M. and by Research grants from Takeda Science Foundation and Uehara Memorial Foundation to T.I.

## Author contributions

K.I. performed all genetic and biochemical experiments (generation of yeast strains and mutants, sucrose density gradients, spot assays), generated the in vitro reconstituted and native cryo-EM samples, processed and interpreted cryo-EM data, N.I. determined the crystal structure of the Otu2 extended OTU domain, R.Bu. and J.C. built and refined the Otu2-40S model, R.Bu. built all models for native Otu2-containing complexes, T.F. performed label-free quantification mass spectrometry analysis, Y.M. performed polyribosome profiles with the eS7-4KR mutant, O.B. and R.Bu. collected cryo-EM data, T.I. co-initiated the project and K.I., T.I., T.B., and R.Be. designed the study and wrote the manuscript with comments from all authors,

## Funding

## Competing interests

The authors declare no competing interests.
