## [Peer Review File · Nature Communications]

Molecular basis for recognition and deubiquitination of 40S ribosomes by Otu2REVIEWER COMMENTS

Reviewer #1 (Remarks to the Author):

Ubiquitination is one of the highly conserved post-modification systems in eukaryotes. Although the best-known function of ubiquitin is to be a tag for proteasomal degradation, ubiquitination has many more functions beyond proteolysis. For example, the ubiquitination-deubiquitination cycle of eS7 is important for efficient translation. eS7 is monoubiquitinated by the ubiquitin ligase Not4 and deubiquitinated by the Otu2 deubiquitinating enzyme. Otu2 binds specifically to the 40S subunit, but it remains unclear how this specific binding of Otu2 to the 40S subunit is possible.

In this manuscript, Ikeuchi and coworkers revealed a structure of Otu2 in a ternary complex with the 40S subunit and ubiquitin. The presented data are clear and informative. We can now speculate how Otu2 binds specifically to the 40S but not to the 80S ribosome. Further, the authors detected recycling factors like ABCE1 and Tma64 as the coimmunoprecipitates with Otu2. These proteins and the structural information will be helpful to prevail the ribosome recycling step in more detail in the future. This reviewer gives the authors some questions out of curiosity and comments on a few minor points for readers to understand this manuscript more precisely.

1. Does the presence or absence of ubiquitin at K83 in eS7 change the conformation of Otu2?
2. Although Otu2 sticks to the 40S ribosome after removing ubiquitin from eS7, can Otu2 bind to the 40S ribosome that is not ubiquitinated from the beginning?
3. Although we can speculate that r-proteins indicate ribosomal proteins, the authors should describe that clearly in the figure legends. The authors also should describe what SSU stands for.
4. It would be better to explain why the authors chose C187 and H300 to make Otu2 mutants at line 124 on page 7.
5. The authors should describe what asterisks indicate in Extended Data Fig. 1b.
6. It is confusing that the numbers of the region for $\alpha 5$ are different between the main text (lines 164-165) and the figure legend in Extended Data Fig. 6.
7. Fig. 2c should be Fig. 3c at lines 249 and 259. Fig. 2b should be Fig. 3b at line 260.
8. There is no description regarding the ternary complex in Extended Data Figs. 8d and 8h. It would also be helpful to indicate where the eIF5 N-terminus is.

Reviewer #2 (Remarks to the Author):

The yeast protein Otu2 has recently been identified as responsible for deubiquitination of eS7, a protein of the small ribosomal subunit. This modification participates in regulating translation efficiency, but the molecular mechanisms underlying the action of Otu2 are poorly characterized. In this manuscript, K. Ikeuchi and colleagues use a combination of cryo-EM, X-Ray crystallography and the newly available Alpha-Fold2 structure prediction software to reveal the 3D structure and positioning of Otu2 on yeast 40S subunit. They also provide a set of biochemical experiments that confirm that Otu2 acts on free 40S subunits and not on the 80S ribosome, as already published by Takehara et al (iScience 2021). They propose a model in which Otu2 would associate to free 40S subunits until formation of the 48S pre-initiation complex, as already formally shown by Takehara and colleagues (iScience 2021). Overall, the various methodologies presented in this study appear sound, and meet the expected standards in the field of biochemistry and structural biology. The noteworthy result brought by this study is the establishment of atomic model of Otu2, bound to a ubiquitinated free 40S ribosomal subunit. However, several points lead to question the overall significance of the work, and fail to convince that this manuscript is worth publishing as it stands, especially in such a high impact journal as Nature Communications.

Major points:

- Abstract boldly states conclusions which can not be directly drawn from the present article: "Here we present biochemical and structural data showing that Otu2 can engage the recycled 40S subunit together with the recycling factors ABCE1 and Tma64 immediately after 60S dissociation for mRNA recycling, and that it dissociates before 48S initiation complex formation." This is only partially true. Indeed, mass spec analysis shows a stronger association of Otu2 with RPS, ABCE1 and Tma64 than Upb3: this does account as biochemical evidences; however nothing on the structural analysis proves this directly, nor adds any information to this observation. As the position of ABCE1 on free 40S subunit is known, and ABCE1 is co-purified with tagged Otu2, have the authors observed some densities in the ABCE1 area in their cryo-EM maps of endogenously purified Otu2(C178S)-40S complexes? This might help understanding the interplay between ABCE1 and Otu2. Similarly, no experiments presented in this manuscript address the role of Otu2 in mRNA recycling. This abstract sentence is misleading (to say the least), and must be modified.

- Introduction: the final paragraph states that "it is not clear at which stage of the translation cycle Otu2 is active and how 40S specificity is achieved". While the second part of the sentence is true, Takehara and colleagues (cited as reference 3) have already clearly demonstrated how Otu2 binds to recycled 40S subunits, 43S but not 48S preinitiation complexes. This certainly narrows down the chronology of action of Otu2. As such, the current study confirms these previously published results, but does not add value to it. As for Otu2 specifically associating to free 40S, this manuscript does not bring a formal proof of the hypothesis proposed (see below). Here again this last sentence should be corrected to state what is truly achieved with the work presented here.

- Role of Otu2 in ribosome biogenesis : mass spec differential analysis shows a stronger association of Otu2 with proteins involved in small ribosomal subunit biogenesis. This indeed suggests a functional role of Otu2 during small ribosomal subunit assembly, but needs to be formally assessed, using for instance Northern blot analyses monitoring the accumulation of pre-rRNA maturation intermediates upon Otu2 deletion. Furthermore, there is a confusion: the deubiquitinating enzyme which has been shown to play a role in human ribosome biogenesis in Montellese et al (2020), cited as reference 35, is USP16, not OTUD6B. The later has not been shown to be involved in this process, and the former has very little similarity with Otu2. This error must be corrected. Furthermore, if the authors do not aim at investigating the role of Otu2 in yeast ribosome biogenesis, this notion remains an hypothesis, which might be worth mentioning in the discussion section, but can definitely not appear as a result of the present study.

-Specificity for 40S binding : the data presented in the article clearly show how its N-terminal part (especially alpha helix $\alpha 4$) and not its OTU domain is specifically required for Otu2 binding to the 40S subunit. The authors observe the Otu2 $\alpha 4$ occupies the same place on the 40S subunit as the C-ter helix of eL19 when the small ribosomal subunit is bound to 60S subunit. The authors jump to the conclusion that N-ter helices and mostly $\alpha 4$ of Otu2 provide the specificity of Otu2 activity for 40S versus 80S ribosomes. While this might very well be true, without experimental check, this remains an hypothesis, not a conclusion that can be drawn from the current data. Since they have almost all the tools at hand, could the authors perform competitive binding assays between the N-ter domain of Otu2 and the C-ter of eL19 to purified 40S subunits, to assess whether Otu2 and eL19 are mutually exclusive? Inversely, can an excess of Otu2 break the eB12 intersubunit bridge within 80S ribosomes? This might help understanding the dynamics of association/dissociation of Otu2 to small ribosomal subunits, and maybe how Otu2 specifically binds to 40S subunits.

Minor points:

Figure 1: The quantitative mass spec analysis is not evident to read, and its legend incomplete. Could the authors be a bit more explicit in the results or the legend? Importantly, what do the grey dots correspond to? What is the rationale for naming one dot and not another (cutoff, biological interest???)
Figure 1a should read yeast strains, not stains.

Otu2 Mutants: can the authors explain on which basis they chose the point mutations C178S and H300A that inactivate Otu2? C178S was already used in Takehara and colleagues, this must be acknowledged; what about H300A?

Establishment of an atomic model for Otu2 is not very easy to follow. Cryo-EM maps might have lacked resolution, and alpha-fold2 remains a prediction, hence the interest to perform X-Ray structure determination. However the sentence “This enabled us to obtain an overall model of eS7-mono-Ub bound Otu2 based on the crystal structure of ubiquitin-bound Otu1” is confusing. Was Otu1 atomic structure used for molecular replacement? It might be interesting to discuss the differences found between alpha-fold2 and the X-Ray structures in more details; but maybe not the cryo-EM derived atomic model, since cryo-EM maps resolution in the Otu region is quite low according to ext. data fig 4.

Reviewer #3 (Remarks to the Author):

In this work Ikeuchi and collaborators describe a structure of the Otu2 deubiquitinating enzyme and 40 S ribosomes by combined AlphaFold2 prediction, X-ray crystallography and cryo-EM. Their structural work provides explanations for the association of Otu2 with 40S only after its dissociation from 60S, supporting the model previously published that it can serve to reset translation after ribosome recycling.

In brief, the authors first purify and compare by mass spectrometry the proteins associated with Otu2-bound versus Ubp3-bound ribosomes. They then monitor eS7 ubiquitination across sucrose gradients in cells expressing no Otu2, wild type Otu2 or catalytically inactive Otu2, confirm that eS7 de-ubiquitination needs catalytically active Otu2, and that wild type polysome levels needs catalytically active Otu2, and that Otu2 associates with 40S ribosomes. They confirm a synthetic growth defect in cells lacking Ubp3 and Otu2 particularly at high temperature. They reconstitute an Otu2-eS7-Ubi-40S complex in vitro by purifying 80S ribosomes, ubiquitinating them with Not4 E3 ligase, E1 and E2 enzymes, and ATP, splitting the subunits with puromycin and high salt sucrose gradient fractionation, and deubiquitinating the 40S with Otu2, and incubating the ribosomes with a 5X molar excess of catalytically inactive Otu2 to form complexes subjected to cryoEM. They also determined the structure of the N-terminally extended Otu2-OUT domain (150-307 amino acids) by X-ray crystallography and predict the structure of the full length Otu2 by AlphaFold. Finally, they compare their structure with endogenously formed complexes of 40S with catalytically inactive otu2-C178S-Flag3 that they affinity purified. They finally express truncation mutants of Otu2 to confirm that the stable recruitment of Otu2 to 40S depends upon its N-terminal α -helices.

Otu2 was recently identified as an enzyme responsible for de-ubiquitinating eS7, a protein ubiquitinated by the Not4 E3 ligase, and shown to be important for translation levels. It was proposed to act at the level of ribosome recycling for translation re-initiation (reference 3). The structure presented in this work nicely supports this model in that it provides structural evidence that Otu2 associates with 40S ribosomes but cannot associate with 80S ribosomes.

The work is well performed and this structure adds to the growing number of ribosomal structures solved aiming to get a global mechanistic understanding of the translation process.

Unfortunately, my feeling is that this manuscript does not go beyond solving the structure and falls short of providing new biological information, so of being a highly relevant manuscript as it stands.

The citation of the literature in the introduction is not precise, and in part misleading as to what has actually been proven concerning eS7 ubiquitination and Otu2. This is to me a major problem for a manuscript, which does not add new experiments to further understand the biological context of Otu2 function. Specifically:

In the first paper where eS7 was identified as a substrate of Not4 it was shown that free 40S subunits do not have ubiquitinated eS7 (reference 2), but I have no knowledge of a study that has subsequently shown that ubiquitination of eS7 actually occurs after initiation as the authors mention, but without a citation. In fact, if Otu2 binds 40S ribosomes, could it not be that this is why there is no ubiquitinated eS7 in free 40S ribosomes, rather than because eS7 gets ubiquitinated after initiation? Such a model is not incompatible with the authors' own findings that Otu2 is associated with ribosome biogenesis factors and that a role in small subunit biogenesis was shown for the human ortholog of Otu2 as mentioned by the authors themselves (reference 35).

It is not adequate for the authors to cite their own previous studies with regard to the fact that Not4 ubiquitinates eS7 (bottom of page 3). This was a discovery in reference 2. Their previous studies (references 1 and 9) were subsequent to this discovery. Maybe the authors would like to cite their studies related to the beginning of the sentence where they indicate that eS7 is an important target for general translation coordination. If this was the goal of the authors, they should place the references to the part of the sentence that they refer to so that readers can find the relevant information. But I anyway question how their previous work suggests a role of eS7 in general translation coordination. One study (reference 9) indicates that eS7 ubiquitination can serve for NGD if the RQT complex is defective, and the other (reference 1) shows that it contributes to Not5 association with polysomes (a finding previously shown in reference 2 also) and instability of non-optimal codon containing mRNAs. Both of these studies focus on mRNA decay not translation coordination.

Regarding translational roles of eS7, the authors do not mention that eS7 ubiquitination prevents protein aggregation even under normal conditions (reference 2), is associated with selective mRNA translation under ER stress (reference 28) and prevents translation of poly arginine codons (doi:10.1016/j.celrep.2021.109633).

If the authors mean to refer to the fact that eS7 is important for presence of Not5 in polysomes and that Not5 has roles in translation, this should be described as such with appropriate references.

POINT-BY-POINT response to the reviewers

Reviewer #1 (Remarks to the Author):

Ubiquitination is one of the highly conserved post-modification systems in eukaryotes. Although the best-known function of ubiquitin is to be a tag for proteasomal degradation, ubiquitination has many more functions beyond proteolysis. For example, the ubiquitination-deubiquitination cycle of eS7 is important for efficient translation. eS7 is monoubiquitinated by the ubiquitin ligase Not4 and deubiquitinated by the Otu2 deubiquitinating enzyme. Otu2 binds specifically to the 40S subunit, but it remains unclear how this specific binding of Otu2 to the 40S subunit is possible.

In this manuscript, Ikeuchi and coworkers revealed a structure of Otu2 in a ternary complex with the 40S subunit and ubiquitin. The presented data are clear and informative. We can now speculate how Otu2 binds specifically to the 40S but not to the 80S ribosome. Further, the authors detected recycling factors like ABCE1 and Tma64 as the coimmunoprecipitates with Otu2. These proteins and the structural information will be helpful to prevail the ribosome recycling step in more detail in the future. This reviewer gives the authors some questions out of curiosity and comments on a few minor points for readers to understand this manuscript more precisely.

1. Does the presence or absence of ubiquitin at K83 in eS7 change the conformation of Otu2?

We answer this question in the revised version of the manuscript, to which we added a shotgun cryo-EM approach (revised Figures 2 and 3 new Supplementary Figs 3, 5 and 7; see in detail below) to obtain an inventory of native Otu2-containing 40S ribosomal particles. Here we find that Otu2 can bind to 40S complexes independent of the eS7 ubiquitination state. We find particles, mainly initiation complexes, with Otu2 bound to mono-ubiquitinated eS7, but also particles, mainly pre-40S, in which ubiquitin is missing. While the OTU domain is stably positioned on 40S with eS7-ubiquitin present, in the pre-40S particles the OTU domain is delocalized, indicating it is flexible when ubiquitin is absent from K83 of eS7.

2. Although Otu2 sticks to the 40S ribosome after removing ubiquitin from eS7, can Otu2 bind to the 40S ribosome that is not ubiquitinated from the beginning?

Response:

To address this question, we generated a mutant yeast strain where all four lysines of eS7 were exchanged with arginines. Ribosomes incorporating this eS7-4KR mutant protein thus cannot be ubiquitinated at any time. To test for Otu2 association with such ribosomes, we performed polyribosome sucrose gradient analysis. Here, we observed that mutant eS7 is not ubiquitinated while FLAG-tagged Otu2 still comigrated with 40S, confirming that Otu2 is able to bind to 40S independent of eS7 ubiquitination.

We added this experiment in the revised Supplementary Fig. 9a and 9b.

3. Although we can speculate that r-proteins indicate ribosomal proteins, the authors should describe that clearly in the figure legends. The authors also should describe what SSU stands for.

We explained „r-proteins“ in the figure legends and introduced abbreviations SSU and LSU in the main text and all figure legends.

4. It would be better to explain why the authors chose C187 and H300 to make Otu2 mutants at line 124 on page 7.

C178, H300 and N302 are the three conserved residues that form the so-called “catalytic triad” present in cysteine peptidases in general and also in most members of the OTU family of deubiquitinases including Otu2’s OTU domain (see for example reviews by Mevissen and Komander, *Ann. Rev. Biochem.*, 2017 or Du et al, *Front. Med.*, 2020). Mutating these residues usually renders these enzymes inactive. For Otu2 C178S, this has been already demonstrated in Takehara et al., *Cell Rep*, 2021. H300A was chosen based on sequence alignments. As suggested, we added a brief explanation in the main text.

5. The authors should describe what asterisks indicate in Extended Data Fig. 1b.

The asterisks indicate mono-ubiquitinated eS7 in the 40S fraction. We added the explanation in legend of revised Supplementary Figure 1b.

6. It is confusing that the numbers of the region for $\alpha 5$ are different between the main text (lines 164-165) and the figure legend in Extended Data Fig. 6.

We thank the referee to point out this inconsistency. $\alpha 5$ is from Leu150-Lys165. We corrected this in the legend of revised figure that is now Supplementary Fig. 1.

7. Fig. 2c should be Fig. 3c at lines 249 and 259. Fig. 2b should be Fig. 3b at line 260.

We thank the reviewer for pointing out this mistake. For the revised version new figures were added and we carefully checked that they are referenced correctly in the text.

8. There is no description regarding the ternary complex in Extended Data Figs. 8d and 8h. It would also be helpful to indicate where the eIF5 N-terminus is.

In the revised manuscript, we removed Extended Figure 8 since we observed Otu2-bound 43S and partial 48S complexes initiation complexes as well as pre-40S particles in our new native samples. We show the eIF5 N-terminal domain now in our partial 48S initiation complex in revised Fig. 2b. We also added a description for the ternary complex (eIF2-GTP-initiator tRNA complex) in the main text and in the figure legend.

Reviewer #2 (Remarks to the Author):

The yeast protein Otu2 has recently been identified as responsible for deubiquitination of eS7, a protein of the small ribosomal subunit. This modification participates in regulating translation efficiency, but the molecular mechanisms underlying the action of Otu2 are poorly characterized. In this manuscript, K. Ikeuchi and colleagues use a combination of cryo-EM, X-Ray crystallography and the newly available Alpha-Fold2 structure prediction software to reveal the 3D structure and positioning of Otu2 on yeast 40S subunit. They also provide a set of biochemical experiments that confirm that Otu2 acts on free 40S subunits and not on the 80S ribosome, as already published by Takehara et al (iScience 2021). They propose a model in which Otu2 would associate to free 40S subunits until formation of the 48S pre-initiation complex, as already formally shown by Takehara and colleagues (iScience 2021). Overall, the various methodologies presented in this study appears sound, and meet the expected standards in the field of biochemistry and structural biology. The noteworthy result brought by this study is the establishment of atomic model of Otu2, bound to a ubiquitinated free 40S ribosomal subunit. However, several points lead to question the overall significance of the work, and fail to convince that this manuscript is worth publishing as it stands, especially in such a high impact journal as Nature Communications.

Major points:

- Abstract boldly states conclusions which cannot be directly drawn from the present article: “Here we present biochemical and structural data showing that Otu2 can engage the recycled 40S subunit together with the recycling factors ABCE1 and Tma64 immediately after 60S dissociation for mRNA recycling, and that it dissociates before 48S initiation complex formation.” This is only partially true. Indeed, mass spec analysis shows a stronger association of Otu2 with RPS, ABCE1 and Tma64 than Ubp3: this does account as biochemical evidences; however nothing on the structural analysis proves this directly, nor adds any information to this observation. As the position of ABCE1 on free 40S subunit is known, and ABCE1 is co-purified with tagged Otu2, have the authors observed some densities in the ABCE1 area in their cryo-EM maps of endogenously purified Otu2(C178S)-40S complexes? This might help understanding the interplay between ABCE1 and Otu2. Similarly, no experiments presented in this manuscript address the role of Otu2 in mRNA recycling. This abstract sentence is misleading (to say the least), and must be modified.

We agree with the referee that this manuscript didn't contain direct structural evidence that Otu2 binds to recycled 40S with recycling factors ABCE1 or Tma64 present. We therefore generated a new sample for which we modified the purification approach to better preserve the association of recycling and pre-initiation factors such as ABCE1, Tma64 and eIF3 as well as ribosome biogenesis factors. Taking into account, that expression of Otu2 leads to stable association with 40S subunits, we skipped further affinity purification steps and only performed density gradient centrifugation of the cell lysate. We then harvested the 40S peak and performed a thorough cryo-EM analysis (shotgun approach). This approach is similar to a previously published study from our lab, investigating ABCE1-bound initiation complex in human and yeast (Kratz et al., *EMBO J.* 2020) and here yields an inventory of Otu2 associated particles.

Our new structures (presented in new Figures 2 and 3 as well Supplementary Figures 3, 5 and 7) indeed show that Otu2 is present on recycled and initiating ribosomes together with ABCE1 and factors forming the 43S and even 48S (pre-)initiation complex. We also found classes containing ABCE1, confirming the Otu2 and ABCE1 can bind to the same 40S. In addition, we found one subclass with extra density in a

position where also mRNA recycling factors Tma20 (MCT-1), Tma22 (DENR) and Tma64 (eIF2D) have been found previously (Supplementary Figure 3)

Interestingly, in our Otu2-bound 43S/48S complexes we also find ribosomal protein eL41 present on the 40S, whereas it is missing in our late pre-40S particles. This protein is only incorporated into the 40S after first subunit joining. Thus, this observation suggests that our 43S particles indeed contain 40S that underwent at least one translation cycle including a subunit recycling event.

Taken together, with our new structures we are able to confirm on a structural level, that Otu2 is present on the 40S during various stages of ribosome recycling (with ABCE1 and eIF3), mRNA/tRNA recycling (with Tma64), pre-initiation (with 43S) and initiation (48S), fully consistent with the mass spec data.

Moreover, we were able to subclassify several Otu2-bound pre-40S ribosomal particles indicating a possible involvement of Otu2 in late 40S biogenesis. We find mainly Otu2-containing late pre-40S classes with factors Tsr1, Dim1, Rio2, Enp1(-Ltv1?), Pno1 and Nob1 (in pre-cleavage state) bound (new Figure 2c). The majority of these pre-40S don't contain fully accommodated uS3, but contrary to previous structures, uS10 and RACK1 are already incorporated.

- Introduction: the final paragraph states that "it is not clear at which stage of the translation cycle Otu2 is active and how 40S specificity is achieved". While the second part of the sentence is true, Takehara and colleagues (cited as reference 3) have already clearly demonstrated how Otu2 binds to recycled 40S subunits, 43S but not 48S preinitiation complexes. This certainly narrows down the chronology of action of Otu2. As such, the current study confirms these previously published results, but does not add value to it. As for Otu2 specifically associating to free 40S, this manuscript does not bring a formal proof of the hypothesis proposed (see below). Here again this last sentence should be corrected to state what is truly achieved with the work presented here.

We thank the reviewer for bringing up this issue. In the revised manuscript, we rewrote the introduction (based also on the comments of reviewer #3) and rephrased the respective sentence. Yet, we feel that our new structures presented in the revised version (see above) are adding a substantial value, because they reveal the molecular mode of physical interaction of Otu2 with 40S during various stages of ribosome recycling and translation initiation. In particular, the finding that Otu2 also associates with 48S complexes was not shown before and adds an additional substrate 40S complex for Otu2.

- Role of Otu2 in ribosome biogenesis: mass spec differential analysis shows a stronger association of Otu2 with proteins involved in small ribosomal subunit biogenesis. This indeed suggests a functional role of Otu2 during small ribosomal subunit assembly, but needs to be formally assessed, using for instance Northern blot analyses monitoring the accumulation of pre-rRNA maturation intermediates upon Otu2 deletion. Furthermore, there is a confusion: the deubiquitinating enzyme which has been shown to play a role in human ribosome biogenesis in Montellese et al (2020), cited as reference 35, is USP16, not OTUD6B. The later has not been shown to be involved in this process, and the former has very little similarity with Otu2. This error must be corrected. Furthermore, if the authors do not aim at investigating the role of Otu2 in yeast ribosome biogenesis, this notion remains an hypothesis, which might be worth mentioning in the discussion section, but can definitely not appear as a result of the present study.

With respect to Otu2 involvement in ribosome biogenesis, our new structures show that Otu2 is bound to late pre-40S assembly intermediates (see above) prior to cleavage of the 3' end of 18S rRNA and prior to full uS3 incorporation. We further observe that the catalytic OTU domain of Otu2 is delocalized in these pre-40S and we biochemically confirmed that eS7 is not ubiquitinated in such complexes (Supplementary Figure 8a and 8b).

Since the structures don't give a hint on the potential role of Otu2 during these late maturation events, we also performed – as suggested by the reviewer – Northern Blot analysis to assess the impact of Otu2 (and Ubp3) deletion as well as several Otu2 mutants on abundance of 18S rRNA. Here, we observed a general reduction of pre-rRNA (20S) and also mature rRNA (18S and 25S), indicating that in absence of the eS7b de-ubiquitination system (Otu2 or Otu2/Ubl3 double deletions) less ribosomes are made or ribosomes are degraded at a higher rate. We added these results in the revised manuscript in new Supplementary Figure 8c. Further experiments, however, that could address the exact role of Otu2 during ribosome biogenesis are in our opinion beyond the scope of this paper and should be subject of future studies.

Regarding the confusion with the Montellese et al (2020) reference, the reviewer is correct that in this manuscript mainly the role of USP16 is characterized. However, in addition it is shown that the human homolog of Otu2, OTUD6B, was co-enriched with USP16 in pulldowns using the kinase-dead version of RIOK1 as bait. In agreement with our findings, this suggests a role of OTUD6B (Otu2) in ribosome biogenesis.

Citation:

„In addition to ribosomal proteins and known 40S trans-acting factors, a number of proteins not previously linked to 40S subunit maturation were found to be strongly enriched on the kinase-dead version of RIOK1, with OTUD6B, G3BP1, and USP16 being the most significant among the top scorers (Figure 1C, Supplementary file 2).“

-Specificity for 40S binding: the data presented in the article clearly show how its N-terminal part (especially alpha helix $\alpha 4$) and not its OTU domain is specifically required for Otu2 binding to the 40S subunit. The authors observe the Otu2 $\alpha 4$ occupies the same place on the 40S subunit as the C-ter helix of eL19 when the small ribosomal subunit is bound to 60S subunit. The authors jump to the conclusion that N-ter helices and mostly $\alpha 4$ of Otu2 provide the specificity of Otu2 activity for 40S versus 80S ribosomes. While this might very well be true, without experimental check, this remains an hypothesis, not a conclusion that can be drawn from the current data. Since they have almost all the tools at hand, could the authors perform competitive binding assays between the N-ter domain of Otu2 and the C-ter of eL19 to purified 40S subunits, to assess whether Otu2 and eL19 are mutually exclusive? Inversely, can an excess of Otu2 break the eB12 intersubunit bridge within 80S ribosomes? This might help understanding the dynamics of association/dissociation of Otu2 to small ribosomal subunits, and maybe how Otu2 specifically binds to 40S subunits.

Response:

To address the role of the Otu2 N-terminal domain for 40S specific, especially by competition with the eL19 binding site at the 40S (ES6e) we tested, whether deletion of the C-terminal helix of eL19 would affect the specificity of Otu2 for 40S. Therefore, we generated yeast strains with 80S ribosomes lacking parts of or the entire C-terminal helix of eL19 (new Supplementary Fig. 9c-9e) and analyzed co-migration of Otu2 with

ribosomes on sucrose density gradients. While monitoring whether a fraction of Otu2 can now to bind to 80S in the absence of the eL19 CTD, we observed that Otu2 was still specifically found exclusively in the 40S but not in the 80S fractions. This indicates that eL19-free eB12 bridge on 80S is not a sufficient determinant for 40S specificity.

We also attempted to perform competitive binding assays as suggested by the reviewer, but these experiments were difficult to interpret due to aggregation of truncated eL19 and thus omitted from the paper. Instead, we tested if Otu2 could play an active role in the subunit dissociation process by either supporting splitting of 80S itself or by acting as an anti-association factor similar to ABCE1 (for the 40S) or eIF6 (for the 60S). Yet, both splitting and re-association assays did not reveal any activity of Otu2 in splitting or preventing re-association of 80S (Supplementary Fig. 9f and 9g). All native Otu2-containing particles are bound to factors that would actively prevent 60S binding (ABCE1 and eIF3c-NTD in case of initiation complexes; Tsr1, Rio2 and Dim1 in case of pre-40S), explaining why Otu2 may not be required to carry such an activity.

Taken together, our new data make a significant contribution to understand the dynamics of Otu2 association with small ribosomal subunits: Otu2 can associate with multiple states of ribosomal particles, either pre-40S or any 40S after ribosome splitting into subunits until the stage of start-codon recognition. Specificity for these 40S is established via the Otu2 NTD that employs both, its N-terminal hairpin ($\alpha1$ - $\alpha2$) and $\alpha4$, respectively, both of which show reduced binding to 40S upon deletion. The Otu2-NTD thereby binds to a site not occupied by other factors and 40S binding is independent on the eS7-ubiquitination state. While our structure implies that the Otu2-NTD alone may be able to compete with 60S binding, our experimental data suggest 60S subunits presence could trigger Otu2 dissociation from empty 40S. We thus speculate that Otu2 is a constitutive 40S binding protein, maybe serving as an additional checkpoint for successful initiation or 40S biogenesis, that only dissociates after initiation or 40S maturation are completed.

Minor points:

Figure 1: The quantitative mass spec analysis is not evident to read, and its legend incomplete. Could the authors be a bit more explicit in the results or the legend? Importantly, what do the grey dots correspond to? What is the rationale for naming one dot and not another (cutoff, biological interest???)
Figure 1a should read yeast strains, not stains.

Response:

We modified the panel displaying the mass spectrometric results and added additional explanation in the main text and in the figure legend to make it easier for the reader to understand and interpret the results. The labeling was done partially based on cutoff and partially based on relevance to the study. We added a Supplemental Table (Supplementary Table 1) containing all identified proteins with the ones displayed as colored dots also color coded in the Table. We hope that this makes the data more transparent and accessible to the reader.

Otu2 Mutants: can the authors explain on which basis they chose the point mutations C178S and H300A that inactivate Otu2? C178S was already used in Takehara and colleagues, this must be acknowledged; what about H300A?

This was also a request by referee #1: C178, H300 and N302 are the three conserved residues that can be predicted to form the so-called “catalytic triad” present in cysteine peptidases and Otu2’s OTU

domain. Mutating these residues usually renders these enzymes inactive. For Otu2 C178S, this has been already demonstrated in Takehara et al., *iScience* 2021, which we acknowledge now by citing it when introducing this mutant in the main text. H300A was chosen based on sequence alignments (see revised Supplementary Figure 2). As requested, we added a brief explanation in the main text.

Establishment of an atomic model for Otu2 is not very easy to follow. Cryo-EM maps might have lacked resolution, and alpha-fold2 remains a prediction, hence the interest to perform X-Ray structure determination. However the sentence “This enabled us to obtain an overall model of eS7-mono-Ub bound Otu2 based on the crystal structure of ubiquitin-bound Otu1” is confusing. Was Otu1 atomic structure used for molecular replacement? It might be interesting to discuss the differences found between alpha-fold2 and the X-Ray structures in more details; but maybe not the cryo-EM derived atomic model, since cryo-EM maps resolution in the Otu region is quite low according to ext. data fig 4.

We apologize for the lack of precision in our description for obtaining the models. Indeed, as outlined by the referee, the crystal structure of Otu1 bound to ubiquitin was used to interpret the medium-resolution cryo-EM density present at eS7, representing the Otu2 (OTU domain) bound to ubiquitin. Now, in our new structures, we can unambiguously identify and position the ubiquitin moiety based on clearly resolved secondary structure and comparison to Otu1 is not needed anymore. To solve the crystal structure of the Otu2 OTU domain, we used the structure of the OtuD3 Otu domain (PDB ID 4BOU)(Mevisen, Hospenthal et al. 2013) (and not the one of Otu1) as the search model, as stated in the methods section. As shown in revised Supplementary Figure 6, our X-ray structure and the Alphafold-2 model don't differ much (RMSD of 0.367) except for the loop regions that interact with K83-monoubiquitinated eS7. These were partly absent (Ser294-Gly298 in loop β 3- β 4) or rearranged in the X-ray structure. Here, our new cryo-EM structures allowed to built a more detailed model of 40S-bound Otu2 at local resolutions ranging from below 3 Å for ribosome-interacting regions to 5-7 Å for the more peripheral regions of the Ub-bound OTU domain (see revised Supplementary Fig. 5). Details on generation of molecular models including differences between the X-ray structure and the Alphafold-2 model are given in the revised Methods part.

Reviewer #3 (Remarks to the Author):

In this work Ikeuchi and collaborators describe a structure of the Otu2 deubiquitinating enzyme and 40 S ribosomes by combined AlphaFold2 prediction, X-ray crystallography and cryo-EM. Their structural work provides explanations for the association of Otu2 with 40S only after its dissociation from 60S, supporting the model previously published that it can serve to reset translation after ribosome recycling.

In brief, the authors first purify and compare by mass spectrometry the proteins associated with Otu2-bound versus Ubp3-bound ribosomes. They then monitor eS7 ubiquitination across sucrose gradients in cells expressing no Otu2, wild type Otu2 or catalytically inactive Otu2, confirm that eS7 deubiquitination needs catalytically active Otu2, and that wild type polysome levels needs catalytically active Otu2, and that Otu2 associates with 40S ribosomes. They confirm a synthetic growth defect in cells lacking Ubp3 and Otu2 particularly at high temperature. They reconstitute an Otu2-eS7-Ubi-40S complex in vitro by purifying 80S ribosomes, ubiquitinating them with Not4 E3 ligase, E1 and E2 enzymes, and ATP, splitting the subunits with puromycin and high salt sucrose gradient fractionation, and deubiquitinating the 40S with Otu2, and incubating the ribosomes with a 5X molar excess of catalytically inactive Otu2 to form complexes subjected to cryoEM. They also determined the structure of the N-terminally extended Otu2-OUT domain (150-307 amino acids) by X-ray crystallography and predict the structure of the full length Otu2 by AlphaFold. Finally, they compare their structure with endogenously formed complexes of 40S with catalytically inactive otu2-C178S-Flag3 that they affinity purified. They finally express truncation mutants of Otu2 to confirm that the stable recruitment of Otu2 to 40S depends upon its N-terminal α -helices.

Otu2 was recently identified as an enzyme responsible for de-ubiquitinating eS7, a protein ubiquitinated by the Not4 E3 ligase, and shown to be important for translation levels. It was proposed to act at the level of ribosome recycling for translation re-initiation (reference 3). The structure presented in this work nicely supports this model in that it provides structural evidence that Otu2 associates with 40S ribosomes but cannot associate with 80S ribosomes.

The work is well performed and this structure adds to the growing number of ribosomal structures solved aiming to get a global mechanistic understanding of the translation process.

Unfortunately, my feeling is that this manuscript does not go beyond solving the structure and falls short of providing new biological information, so of being a highly relevant manuscript as it stands.

The citation of the literature in the introduction is not precise, and in part misleading as to what has actually been proven concerning eS7 ubiquitination and Otu2. This is to me a major problem for a manuscript, which does not add new experiments to further understand the biological context of Otu2 function. Specifically:

In the first paper where eS7 was identified as a substrate of Not4 it was shown that free 40S subunits do not have ubiquitinated eS7 (reference 2), but I have no knowledge of a study that has subsequently shown that ubiquitination of eS7 actually occurs after initiation as the authors mention, but without a citation. In fact, if Otu2 binds 40S ribosomes, could it not be that this is why there is no ubiquitinated eS7 in free 40S ribosomes, rather than because eS7 gets ubiquitinated after initiation? Such a model is not incompatible with the authors' own findings that Otu2 is associated with ribosome biogenesis

factors and that a role in small subunit biogenesis was shown for the human ortholog of Otu2 as mentioned by the authors themselves (reference 35).

Response:

We agree with the reviewer that our wording with respect to literature and citation was somewhat unfortunate. We reworked the introduction accordingly.

We agree with the reviewer that there is no study so far that directly shows that eS7-monoubiquitination occurs after initiation. We drew this conclusion based on the fact that eS7-mono ubiquitination was found mainly on 80S and actively translating polysomes, indicating that the modification is added at least after subunit joining. We modified the revised introduction with respect to eS7 ubiquitination accordingly.

We also agree with the reviewers' explanation for free 40S not being ubiquitinated. Our new cryo-EM data clearly show the pre-40S assembly intermediates are not mono-ubiquitinated at eS7 and confirm this with a native pullout of pre-40S via tagged Rio2 (new Supplementary Fig. 8). We show that Otu2 can in principle bind any kind of known 40S ribosomal particles, since its binding site does not overlap with any other known 40S ribosome binding factor. We thus incorporated the reviewers' comment as part of our revised discussion.

It is not adequate for the authors to cite their own previous studies with regard to the fact that Not4 ubiquitinates eS7 (bottom of page 3). This was a discovery in reference 2. Their previous studies (references 1 and 9) were subsequent to this discovery. Maybe the authors would like to cite their studies related to the beginning of the sentence where they indicate that eS7 is an important target for general translation coordination. If this was the goal of the authors, they should place the references to the part of the sentence that they refer to so that readers can find the relevant information.

Response:

We agree and removed references 1 and 9 when mentioning the fact that Not4 ubiquitinates eS7, and cited our work at more appropriate places.

But I anyway question how their previous work suggests a role of eS7 in general translation coordination. One study (reference 9) indicates that eS7 ubiquitination can serve for NGD if the RQT complex is defective, and the other (reference 1) shows that it contributes to Not5 association with polysomes (a finding previously shown in reference 2 also) and instability of non-optimal codon containing mRNAs. Both of these studies focus on mRNA decay not translation coordination.

Response:

We agree and removed the term „general translation coordination“ in our revised manuscript and cited refs 1 and 9 with more appropriate wording only in the context of the activity mentioned by the reviewer.

Regarding translational roles of eS7, the authors do not mention that eS7 ubiquitination prevents protein aggregation even under normal conditions (reference 2), is associated with selective mRNA

translation under ER stress (reference 28) and prevents translation of poly arginine codons (doi:10.1016/j.celrep.2021.109633).

Response:

We now mention these findings with the appropriate references.

If the authors mean to refer to the fact that eS7 is important for presence of Not5 in polysomes and that Not5 has roles in translation, this should be described as such with appropriate references.

Response:

Our intention to attribute a role of eS7-ubiquitination for “general translation coordination” was exactly this: the fact that eS7-ubiquitination is required to enable a readout for codon optimality by Ccr4-NOT during canonical translation made us claim a role for general translation coordination. We understand that the reviewer may consider this wording not entirely appropriate. Therefore, in the revised manuscript, we removed the phrase and explicitly describe fact that eS7 is important for presence of Not5 in polysomes and that Not5 has roles in translation-coupled RNA decay with appropriate references, as requested.

REVIEWERS' COMMENTS

Reviewer #1 (Remarks to the Author):

The revised manuscript by Ikeuchi and coworkers is substantially improved over the original manuscript. The authors made a serious effort to address the criticisms that were made during the first review. I am satisfied with all responses to my comments. Consequently, the manuscript now seems more appropriate to consider for publication.

Reviewer #2 (Remarks to the Author):

Ikeuchi and colleagues thoroughly and extensively revised their initial manuscript, and took into account all the suggestions and comments proposed by the three reviewers. This huge structural and functional work that was added to their already substantial initial results greatly improves the quality and the significance of their study on the role of deubiquitination of eS7 by Otu2 at various life stages of ribosomes. I also would like to thank the authors to have elegantly corrected a remark that I wrongly emitted about the putative role of de-ubiquitinase OTU6DB in ribosome biogenesis in reference nb 37 (Montellese et al, 2020), which I totally missed. This work is now, in my opinion, original and significant, and totally supports its conclusions and claims. This manuscript is thus perfectly fit to be published in a journal such as Nature Communications.

Reviewer #3 (Remarks to the Author):

In their revised manuscript, Ikeuchi et al., have addressed most specific issues raised by the 3 reviewers.

Concerning my comments about referencing the literature, while they have improved in particular references to previous work as I suggested, there remains over-statements which must be changed, because otherwise future authors are going to refer to these statements that in fact have not been proven experimentally.

On page 3: "eS7 ubiquitination occur on 80S ribosomes »: I have gone through the cited manuscript (reference 27) and could not find an experiment that shows this. In the methods of that manuscript

there is a deubiquitination experiment, but no ubiquitination experiment. Instead, what it is seen in this cited manuscript, and was already noted in the first paper on Rps7A ubiquitination by Not4 (Panasenko 2012), is that Rps7 is only observed ubiquitinated in 80S fractions and polysomes, but not in free 40S fractions. My comment about the fact that since Otu2 binds 40S, this could explain why no ubiquitinated eS7 is detected on the different kinds of 40S particles. It is not necessarily that ubiquitination occurs after subunit joining. I believe the authors need to be very careful about not overstating what the data shows.

Alternatively, the authors can do the experiment: in this manuscript they have a ubiquitination assay, namely that of 80S by Not4. Does this work with 40S? Have they tested this? Maybe they have the experiment.

I remain a little skeptical that the structures provided in this manuscript reveal the “first mechanistic insights into the Otu2-driven deubiquitination for translation reset....” As mentioned in the abstract. I agree that this manuscript describes the molecular basis for recognition of eS7 with K63-linked ubiquitination, for why Otu2 does not bind 80S ribosomes, and it shows that Otu2 can generally bind 40S particles. However, I feel that this manuscript falls somewhat short of providing new biological information.

Point-by-point answers

Reviewer #3 (Remarks to the Author):

In their revised manuscript, Ikeuchi et al., have addressed most specific issues raised by the 3 reviewers.

Concerning my comments about referencing the literature, while they have improved in particular references to previous work as I suggested, there remains overstatements which must be changed, because otherwise future authors are going to refer to these statements that in fact have not been proven experimentally.

On page 3: “eS7 ubiquitination occur on 80S ribosomes »: I have gone through the cited manuscript (reference 27) and could not find an experiment that shows this. In the methods of that manuscript there is a deubiquitination experiment, but no ubiquitination experiment. Instead, what it is seen in this cited manuscript, and was already noted in the first paper on Rps7A ubiquitination by Not4 (Panassenko 2012), is that Rps7 is only observed ubiquitinated in 80S fractions and polysomes, but not in free 40S fractions. My comment about the fact that since Otu2 binds 40S, this could explain why no ubiquitinated eS7 is detected on the different kinds of 40S particles. It is not necessarily that ubiquitination occurs after subunit joining. I believe the authors need to be very careful about not overstating what the data shows.

Answer:

As suggested by the reviewer we changed the mentioned sentence in order to avoid overstating any data. Since we can indeed not say when exactly the eS7 ubiquitylation by Not4 occurs, also on the 40S or only on the 80S, it reads now on page 3:

“Later studies suggest that ubiquitinated eS7 is present only on 80S ribosomes...”

Alternatively, the authors can do the experiment: in this manuscript they have a ubiquitination assay, namely that of 80S by Not4. Does this work with 40S? Have they tested this? Maybe they have the experiment.

Answer:

We haven't done this experiment. However, it is unlikely to be very informative since it has been shown by Panassenko et al., 2012, that *in vitro* Not4 is capable of efficiently ubiquitylating purified recombinant eS7a. Therefore, it is likely that *in vitro* Not4 can ubiquitylate eS7a independent of its context.

Therefore, we would prefer to refrain from making any strong statement about this issue in agreement with the reviewer's comment.

I remain a little skeptical that the structures provided in this manuscript reveal the "first mechanistic insights into the Otu2-driven deubiquitination for translation reset..." As mentioned in the abstract. I agree that this manuscript describes the molecular basis for recognition of eS7 with K63-linked ubiquitination, for why Otu2 does not bind 80S ribosomes, and it shows that Otu2 can generally bind 40S particles. However, I feel that this manuscript falls somewhat short of providing new biological information.

Answer:

We agree that this sentence should be adjusted and accordingly we omitted the 'first' claim. However, we still feel that our work reveals previously unknown and valuable mechanistic insights into the Otu2 activity on the ribosome which is of biological relevance.